# Structural insights into lipid chain-length selectivity and allosteric regulation of FFA2

Mai Kugawa[1,2,14], Kouki Kawakami [1,14], Ryoji Kise[3,14], Carl-Mikael Suomivuori [4,5,6,7], Masaki Tsujimura [8], Kazuhiro Kobayashi [1], Asato Kojima[1,9], Wakana J. Inoue[9], Masahiro Fukuda [1], Toshiki E. Matsui[1,9], Ayami Fukunaga[1,2], Junki Koyanagi[9], Suhyang Kim[1], Hisako Ikeda[1], Keitaro Yamashita [1], Keisuke Saito [1,10], Hiroshi Ishikita [1,10], Ron O. Dror [4,5,6,7], Asuka Inoue [3,11,12] ✉ & Hideaki E. Kato [1,2,9,12,13] ✉

The free fatty acid receptor 2 (FFA2) is a G protein-coupled receptor (GPCR) that selectively recognizes short-chain fatty acids to regulate metabolic and immune functions. As a promising therapeutic target, FFA2 has been the focus of intensive development of synthetic ligands. However, the mechanisms by which endogenous and synthetic ligands modulate FFA2 activity remain unclear. Here, we present the structures of the human FFA2–Gi complex activated by the synthetic orthosteric agonist TUG-1375 and the positive allosteric modulator/allosteric agonist 4-CMTB, along with the structure of the inactive FFA2 bound to the antagonist GLPG0974. Structural comparisons with FFA1 and mutational studies reveal how FFA2 selects specific fatty acid chain lengths. Moreover, our structures reveal that GLPG0974 functions as an allosteric antagonist by binding adjacent to the orthosteric pocket to block agonist binding, whereas 4-CMTB binds the outer surface of transmembrane helices 6 and 7 to directly activate the receptor. Supported by computational and functional studies, these insights illuminate diverse mechanisms of ligand action, paving the way for precise GPCR-targeted drug design.

Lipids are versatile biomolecules that serve not only as energy reservoirs and structural components of membranes but also as essential signaling mediators for maintaining homeostasis, particularly through G protein-coupled receptors (GPCRs). Lipid mediators that act as GPCR ligands include prostaglandins, leukotrienes, lysophospholipids, endocannabinoids, oxidized lipids, and free fatty acids (FFAs). These lipid ligands interact with and activate one or more of the approximately 40 lipid GPCRs in humans, regulating diverse physiological functions from bone and vasculature development to modulation of nociception, immunomodulation, synaptic plasticity, and metabolic homeostasis[1].

Most lipid ligands typically comprise a fatty acid chain and a polar group, exhibiting chemical diversity due to variations in chain length,

[1]Research Center for Advanced Science and Technology, The University of Tokyo, Meguro, Tokyo, Japan. [2]Department of Biological Sciences, Graduate School of Science, The University of Tokyo, Bunkyo, Tokyo, Japan. [3]Graduate School of Pharmaceutical Sciences, Tohoku University, Sendai, Japan. [4]Department of Computer Science, Stanford University, Stanford, CA, USA. [5]Department of Molecular and Cellular Physiology, Stanford University School of Medicine, Stanford, CA, USA. [6]Department of Structural Biology, Stanford University School of Medicine, Stanford, CA, USA. [7]Institute for Computational and Mathematical Engineering, Stanford University, Stanford, CA, USA. [8]Department of Advanced Interdisciplinary Studies, The University of Tokyo, Meguro, Tokyo, Japan. [9]Department of Life Sciences, School of Arts and Sciences, The University of Tokyo, Meguro, Tokyo, Japan. [10]Department of Applied Chemistry, The University of Tokyo, Bunkyo, Tokyo, Japan. [11]Graduate School of Pharmaceutical Sciences, Kyoto University, Kyoto, Japan. [12]FOREST, Japan Science and Technology Agency, Kawaguchi, Saitama, Japan. [13]CREST, Japan Science and Technology Agency, Kawaguchi, Saitama, Japan. [14]These authors contributed equally: Mai Kugawa, Kouki Kawakami, Ryoji Kise. ✉e-mail: iaska@tohoku.ac.jp; c-hekato@g.ecc.u-tokyo.ac.jp

saturation, oxidation status, and polar group composition. Different lipid ligands are recognized by specific receptor groups, with FFA receptors being a particularly interesting group as they are the only receptors that respond to lipids with markedly different chain lengths. In humans, there are four FFA receptors (FFA1-FFA4); FFA1 and FFA4 specifically recognize long-chain fatty acids (LCFAs) with aliphatic tails of more than 14 carbons, whereas FFA2 and FFA3 are responsive to short-chain fatty acids (SCFAs) with aliphatic tails of 2–6 carbons.

Among these FFA receptors, FFA2 has emerged as a pivotal GPCR with profound physiological significance. Expressed across various cell types, including immune cells, adipose tissue, intestinal epithelial and endocrine cells, and pancreatic β-cells, FFA2 plays a crucial role in regulating immune function[2], lipid metabolism[3,4], and the secretion of glucagon-like peptide 1 (GLP-1) and insulin through the activation of the Gi or Gq pathways[5–7]. Consequently, disruption of FFA2 signaling has been implicated in several inflammatory and metabolic diseases such as ulcerative colitis, obesity, and diabetes, leading to the development of numerous non-lipid synthetic agonists and antagonists targeting FFA2[8]. For example, one of the most recently synthesized and reported FFA2 agonists is TUG-1375[9]. This FFA2-selective orthosteric agonist exhibits higher potency, enhanced water solubility, and improved chemical stability compared to SCFAs[9], and induces migration of neutrophils, and inhibits lipolysis in adipocytes[9]. Another notable compound is 4-CMTB, also known as phenylacetamide 1 or AMG7703, which was the first described FFA2-selective synthetic activator and has been used to explore the physiology and pathophysiology of FFA2[8,10], 4-CMTB has attracted attention due to its unique pharmacological properties – it functions as both an allosteric agonist as well as a positive allosteric modulator (PAM)[11,12] – and its beneficial effects on allergic asthma[13] and dermatitis[14]. Among several FFA2 antagonists, GLPG0974 is the best characterized one and has various biological effects, including inhibition of neutrophil chemotaxis[15,16]. Importantly, GLPG0974 went to a clinical trial, and its effectiveness and safety have been established in a phase 2 trial for treating ulcerative colitis[17]. However, despite the development of numerous FFA2-selective agonists and antagonists, the precise mechanisms by which these compounds interact with and activate or inactivate FFA2 remain elusive.

Here, we determine the cryo-electron microscopic (cryo-EM) structures of the human FFA2–Gi signaling complex simultaneously bound to TUG-1375 and 4-CMTB, as well as inactive FFA2 bound to GLPG0974. Combined with computational and molecular pharmacological analyses, the structural information provides important mechanistic insights into SCFA selection and receptor activation/inactivation by orthosteric agonists, allosteric agonists, and antagonists.

## Results

### Structure determination of FFA2 in the active and inactive states
To improve the expression level of human FFA2, we truncated the C-terminal five residues after S325 (Supplementary Fig. 1a, b). For the structural studies of the active FFA2–Gi signaling complex bound to TUG-1375 and 4-CMTB, we expressed the truncated FFA2 construct in HEK293S cells and purified it in the presence of the orthosteric agonist TUG-1375 and the allosteric agonist 4-CMTB. Concurrently, we expressed and purified a wild-type (WT) human Gi heterotrimer ($G\alpha_{i1}\beta_1\gamma_2$) and the single-chain variable fragment scFv16, as described previously[18,19]. We then incubated the purified receptor with the Gi heterotrimer and scFv16 and further purified the reconstructed complex by gel-filtration chromatography (Supplementary Fig. 1c). The prepared complex was vitrified, imaged using a Titan Krios cryo-electron microscope, and the structure was determined at a nominal resolution of 3.19 Å (Fig. 1a, b and Supplementary Fig. 1d–h). The relatively high-resolution density map allowed for the accurate modeling of most residues of FFA2, the Gi heterotrimer, and the TUG-1375 and 4-CMTB ligands (Fig. 1a, b and Supplementary Fig. 1i–m), whereas 45 residues at the helix 8, the C-terminus (R281-S325), and 10 residues in the extracellular loop 2 (ECL2, T152-E161) of the receptor were not resolved, suggesting their highly dynamic properties.

To facilitate the determination of the antagonist-bound FFA2 structure through cryo-EM analysis, we generated the chimeric FFA2-BRIL protein by inserting cytochrome b562 RIL (BRIL) into the intracellular loop 3 (ICL3) of FFA2[20] (Supplementary Fig. 2a). We then purified FFA2-BRIL in the presence of the antagonist GLPG0974 and coupled it with the anti-BRIL antibody BAG2[21]. The resulting complex was further purified by gel-filtration chromatography (Supplementary Fig. 2b), vitrified, imaged, and the structure was determined at a nominal resolution of 3.36 Å (Fig. 1c, d and Supplementary Fig. 2c–g). The density map allowed for the tracing of residues 4-291 of the receptor, except for seven residues in ECL2 (N151-R157) and three in ICL3 (Q209-L211), with additional density suitably positioned for the antagonist GLPG0974 (Fig. 1c, d and Supplementary Fig. 2h–j). We note that the density of the helix 8 of the receptor appears stronger in this structure compared to that in the FFA2–Gi complex, allowing for the modeling of the region (Supplementary Fig. 2h).

### Recognition of orthosteric agonists by FFA2
Chemically, TUG-1375 is composed of carboxyl, thiazolidine, 2-chlorophenyl, benzoyl, and dimethylisoxazole groups (Fig. 2a) and selectively activates FFA2 with sub-micromolar affinity[9]. In the structure of the FFA2–Gi complex, TUG-1375 occupies an extracellular orthosteric pocket and interacts with several residues from transmembrane helices (TMs) 3–7 (Fig. 2b–d). The carboxyl group of TUG-1375 forms a salt

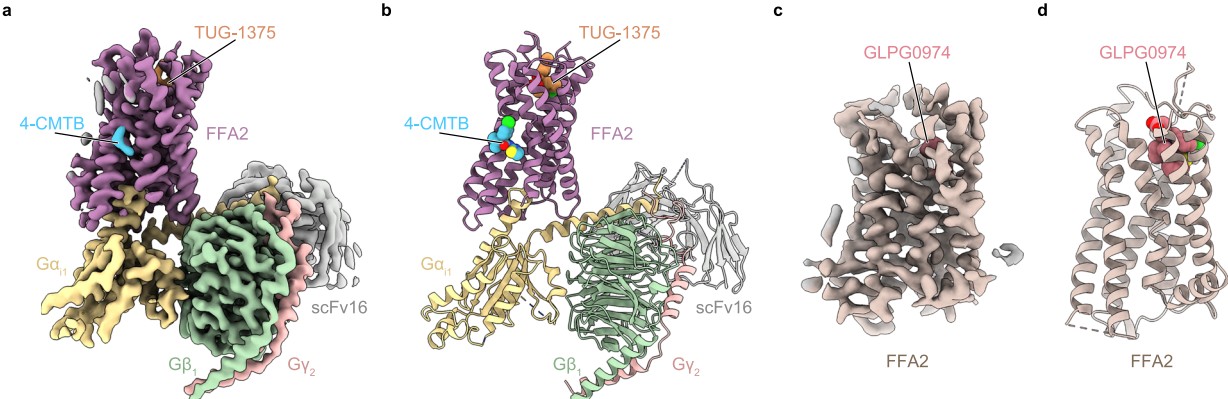

**Fig. 1 | Overall structures of FFA2 in the active and inactive states. a, b** Cryo-EM density map (**a**) and model (**b**) of the FFA2–Gi1 complex bound to TUG-1375 and 4-CMTB. **c, d** Cryo-EM density map (**c**) and model (**d**) of FFA2-BRIL bound to GLPG0974.

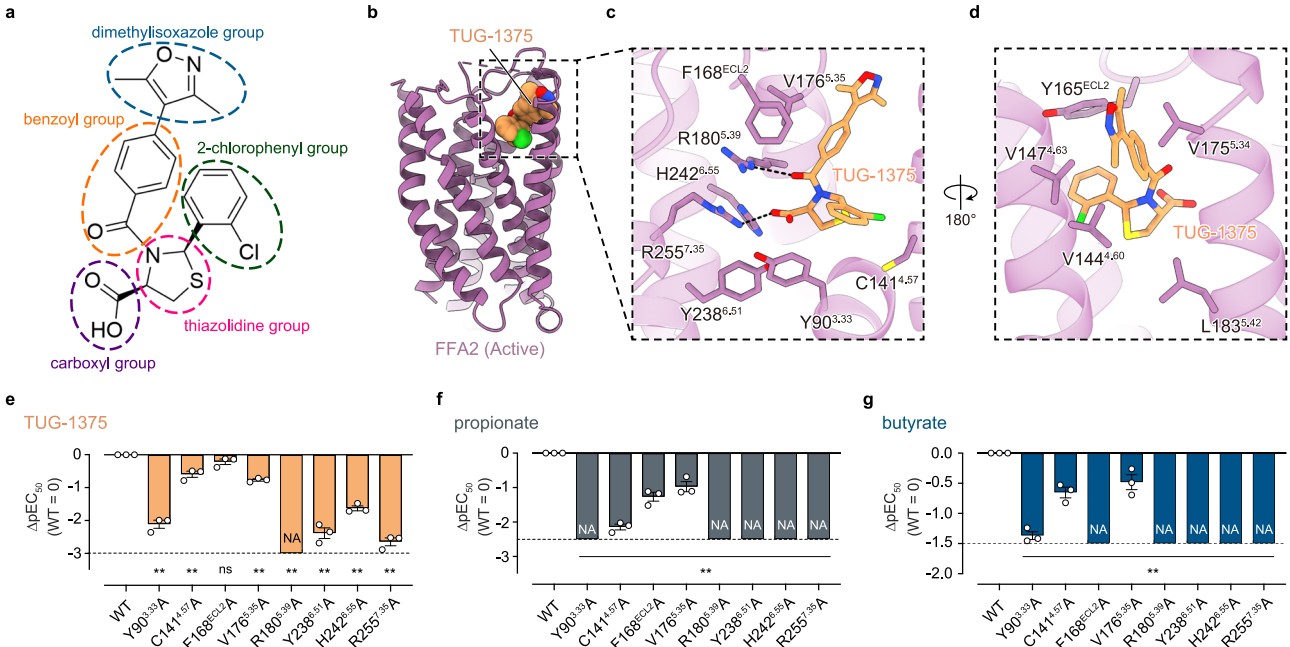

**Fig. 2 | Characterization of the orthosteric ligand pocket. a** Chemical structure of TUG-1375. **b–d** Overall structure of FFA2 bound to TUG-1375 (**b**), and enlarged views of the ligand pocket focused on TUG-1375 (**c, d**). Amino acid numbering and Ballesteros–Weinstein numbering[22] for FFA2 are indicated. **e–g** Mutagenesis analysis of the TUG-1375-binding site using the TGFα shedding assay. The effect on the potency of TUG-1375 (**e**), propionate (**f**), and butyrate (**g**) was evaluated by $pEC_{50}$ values. The $pEC_{50}$ values were normalized by that of WT with similar expression levels ($\Delta pEC_{50}$ values) (See also Supplementary Fig. 3a). NA represents parameters not available owing to a lack of ligand response. Bars and error bars represent the mean and SEM, respectively, of three independent experiments, each performed in duplicate. ** represents $p < 0.01$ with one-way ANOVA followed by Dunnett's test for multiple comparison analysis with reference to WT. ns, not significantly different between the groups. See also the Source Data file for additional statistics and exact $p$-values.

bridge with $R255^{7.35}$ (superscripts denote Ballesteros–Weinstein numbering[22]), polar interactions with $R180^{5.39}$, $Y238^{6.51}$, and $H242^{6.55}$, and an anion·π interaction[23] with $Y90^{3.33}$. In addition, the benzoyl group forms a hydrogen bond with $R180^{5.39}$ and hydrophobic interactions with $V144^{4.60}$, $F168^{ECL2}$, and $V176^{5.35}$, while the thiazolidine, 2-chlorophenyl, and dimethylisoxazole groups form extensive hydrophobic interactions with $L183^{5.42}$, $C141^{4.57}/Y165^{ECL2}$, and $V147^{4.63}/V175^{5.34}$, respectively (Fig. 2c, d).

To analyze the functional contribution of these residues, we generated eight single-alanine-substituted mutants and evaluated their signal activity upon TUG-1375 stimulation. As a control experiment, we first assessed cell-surface expression levels of WT FFA2 and these mutants by flow cytometry analysis (Supplementary Fig. 3a). We then measured G-protein signaling, normalized by their expression levels, using a TGFα shedding assay. In all mutants except for $F168^{ECL2}$A, signaling activities were attenuated more than 3-fold (as assessed by $EC_{50}$ values) (Fig. 2e and Supplementary Fig. 3b, c), indicating that most observed interactions contribute to FFA2 activation. Notably, the mutations in the polar residues that interact with the carboxyl group of TUG-1375 ($Y90^{3.33}$A, $R180^{5.39}$A, $Y238^{6.51}$A, $H242^{6.55}$A, and $R255^{7.35}$A) resulted in reduction of signal activity by over 100-fold, underscoring the significance of these interactions between the carboxyl group of TUG-1375 and the polar residues of FFA2.

Next, we analyzed the binding mode of endogenous SCFA ligands. Given the shared carboxyl group between TUG-1375 and SCFAs, we hypothesized that the carboxyl group of SCFAs similarly binds to the polar residues ($Y90^{3.33}$, $R180^{5.39}$, $Y238^{6.51}$, $H242^{6.55}$, and $R255^{7.35}$). To test this hypothesis, we evaluated the signaling activity of the mutants upon stimulation with the SCFAs propionate and butyrate, observing a decrease in signaling activity by over 100-fold in the polar residue mutants (Fig. 2f, g and Supplementary Fig. 3b, c). These results demonstrate the essential role of these polar residues in FFA2 activation by SCFAs as well.

Interestingly, despite the smaller size of SCFAs compared to TUG-1375, the signaling activity was attenuated in all mutants of TUG-1375 binding sites we created, leading to an apparent inconsistency, as $F168^{ECL2}$ and $V176^{5.35}$ are not expected to interact with the SCFAs. Thus, to further analyze the contribution of these residues to SCFA binding, we performed molecular dynamics (MD) simulations by modeling FFA2 bound to propionate (Methods; Supplementary Fig. 4). We conducted 12 independent simulations and found that propionate is quite dynamic (Supplementary Fig. 4a). In 4 simulations, propionate stably interacts with $Y90^{3.33}$, $R180^{5.39}$, or $R255^{7.35}$ (Supplementary Fig. 4b). However, in the remaining simulations, propionate loses these interactions and transiently interacts with other surrounding residues, including $F168^{ECL2}$ and $V176^{5.35}$ (Supplementary Fig. 4c). In one of these simulations, propionate is released into the bulk solvent (Supplementary Fig. 4d). Combined with the mutagenesis data, these results suggest that the efficient activation of the receptor by SCFA ligands is achieved not only by the most stable interactions in the orthosteric binding pocket, but also by more transient interactions associated with alternative binding poses.

## Chain-length selectivity by FFA2

Next, we focused on the chain length selectivity of FFA2. FFA2 selectively recognizes fatty acids whose aliphatic chain has fewer than 6 carbons, but the structural basis for this selectivity has remained elusive. To understand why FFA2 cannot accept longer fatty acids, we compared our TUG-1375-bound FFA2–Gi structure with the previously reported structure of FFA1–GsqiN bound to the synthetic agonist TAK-875[24], because FFA1 is phylogenetically close to FFA2 but selectively recognizes LCFAs instead of SCFAs. The structural comparison first revealed that the carboxyl groups of both TUG-1375 and TAK-875 share similar binding modes (Fig. 3a–c). Both FFA1 and FFA2 have a shallow and narrow orthosteric binding pocket, and the

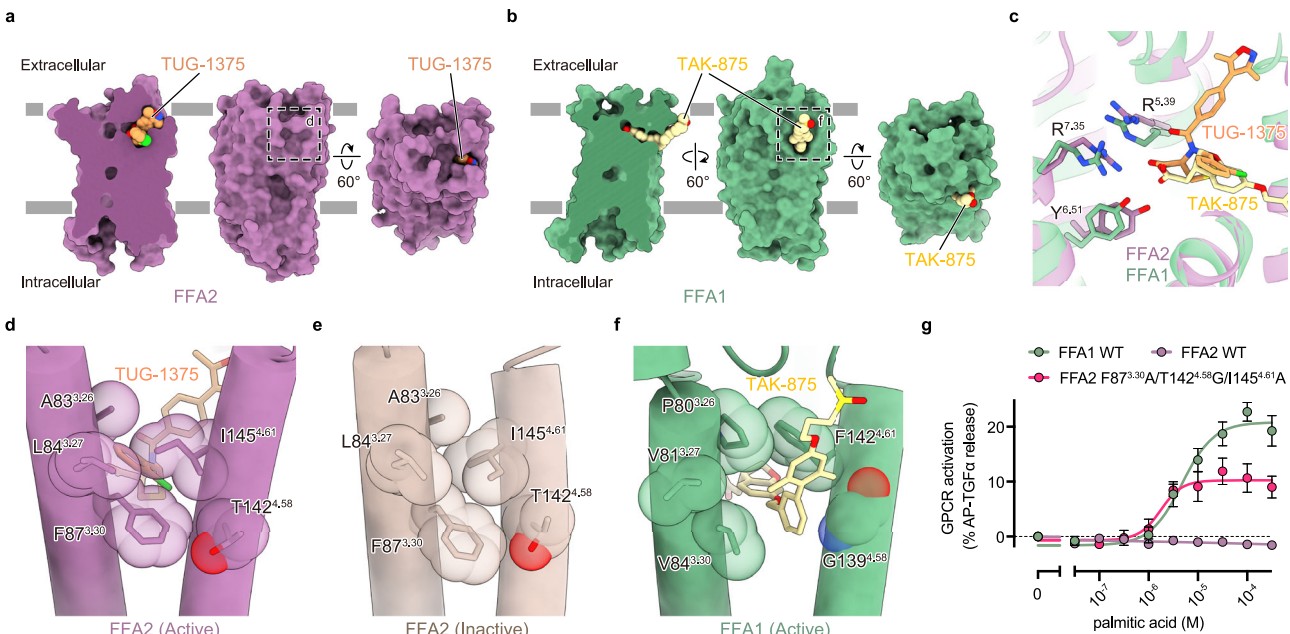

**Fig. 3 | Lipid chain length selectivity. a, b** Cross-section (left) and surface representation (center and right) of FFA2 (**a**) and FFA1[24] (PDB ID: 8EJC) (**b**). **c** Superimposed images of FFA2 bound to TUG-1375 (purple, orange) and FFA1 bound to TAK-875 (green, yellow) focused on the orthosteric site. **d–f** Structural comparison around FFA1's ligand entry site among active FFA2 (**d**), inactive FFA2 (**e**), and active FFA1 (**f**). **g** Mutagenesis analysis of the FFA2 mutant with a ligand entry site mimicking that of FFA1 (F87$^{3.30}$A/T142$^{4.58}$G/I145$^{4.61}$A, magenta) using the TGFα shedding assay upon stimulation with palmitic acid (a long-chain fatty acid). Symbols and error bars represent the mean and SEM, respectively, of 3 independent experiments, each performed in duplicate.

carboxyl groups of both ligands are attached to the bottom of the pocket via interactions with two conserved Arg residues (R$^{5.39}$ and R$^{7.35}$) and one Tyr residue (Y$^{6.51}$) (Fig. 3c). Given that mutations of these Arg residues abolish the receptor activities for both FFA1[25] and FFA2 (Fig. 2e–g), this suggests that the initial critical step for receptor activation (i.e., the recognition of the ligand's carboxyl group by Arg residues) is conserved between FFA1 and FFA2.

The carboxyl groups of TUG-1375 and TAK-875 bind to similar positions in FFA2 and FFA1, respectively. However, the remaining moieties of these ligands extend in significantly different directions. TUG-1375 extends towards the extracellular side and is exposed to the bulk solvent, while TAK-875 extends parallel to the membrane and is exposed to the lipid bilayer (Fig. 3a, b).

The difference in the position of the pore opening may explain why LCFAs cannot activate FFA2. LCFAs are hydrophobic ligands due to their long aliphatic tails and are typically stable in the lipid membrane, accessing the ligand binding pocket from the lateral side of the receptor. However, FFA2's pore opens to the extracellular solvent; thus, to bind to the pocket of FFA2, LCFAs must leave the membrane and access the binding pocket from the extracellular side, which is energetically unfavorable. Moreover, the orthosteric binding pockets of FFA1 and FFA2 are both shallow and narrow, unable to fully accommodate the long aliphatic tails of LCFAs. Consequently, when LCFAs bind to the receptor, a portion of the hydrophobic tail must protrude from the pocket. Therefore, even if LCFAs manage to enter the pocket of FFA2, they cannot bind stably due to the hydrophobic/hydrophilic mismatch between the lipid carbon chain and the bulk solvent. Based on these considerations, we hypothesized that the lack of a ligand entry site at an appropriate position prevents FFA2 from accepting LCFAs.

To test this hypothesis, we engineered the ligand entry site in the middle of the transmembrane helices of FFA2 (Fig. 3d–g). In FFA1, a wide opening between TMs 3 and 4 is formed by five hydrophobic residues (P80$^{3.26}$, V81$^{3.27}$, V84$^{3.30}$, G139$^{4.58}$, and F142$^{4.61}$) (Fig. 3f). In contrast, these five residues are replaced by A83$^{3.26}$, L84$^{3.27}$, F87$^{3.30}$, T142$^{4.58}$,

and I145$^{4.61}$, respectively, and four of them (L84$^{3.27}$, F87$^{3.30}$, T142$^{4.58}$, and I145$^{4.61}$) form a tight closure in FFA2 (in both TUG-1375-bound active and GLPG0974-bound inactive states) (Fig. 3d, e). To loosen the closure formed by L84$^{3.27}$, F87$^{3.30}$, T142$^{4.58}$, and I145$^{4.61}$, we introduced alanine or glycine mutations to these residues and evaluated the receptor's activity upon stimulation with palmitic acid, a representative LCFA with a 16-carbon chain. Strikingly, unlike FFA2 WT, the triple mutant (F87$^{3.30}$A/T142$^{4.58}$G/I145$^{4.61}$A) was effectively activated by palmitic acid, indicating that the position of the ligand entry site plays a critical role in the chain-length selectivity of FFA2 (Fig. 3g and Supplementary Fig. 3a).

## Allosteric inhibition of FFA2 by GLPG0974

GLPG0974, an FFA2 antagonist developed in 2014, is the only FFA2 ligand to date that has been employed in clinical studies[15,17] (Fig. 4a). Intriguingly, while GLPG0974 is characterized as an orthosteric antagonist[8,26], its precise binding site remains elusive. Although GLPG0974 binding competes with orthosteric agonists such as propionate, no single point mutation in the orthosteric binding pocket severely compromises the binding of GLPG0974[16]. To gain insights into the mechanism by which this antagonist is recognized and inhibits the receptor's activity, we focused on the GLPG0974-bound FFA2 structure (Fig. 4).

Despite the presence of clear density for GLPG0974 bound to the receptor, there was ambiguity regarding its orientation due to the limited resolution (Fig. 1c and Supplementary Figs. 2i, 5a, 5b). While the benzothiophene-3-carbonyl group unambiguously fits into the density at the bottom of the binding pocket, there remain two possibilities for the fitting of the 3-chloro-benzyl amino and carboxyl groups (poses 1 and 2), with pose 1 fitting better into the density. To further verify the binding mode, we performed MD simulations (Methods) and found that in three rounds of simulations, GLPG0974 in pose 1 consistently binds more stably to the pocket compared to pose 2 (Supplementary Fig. 5c–j). Therefore, we used pose 1 of GLPG0974 for further structural analysis.

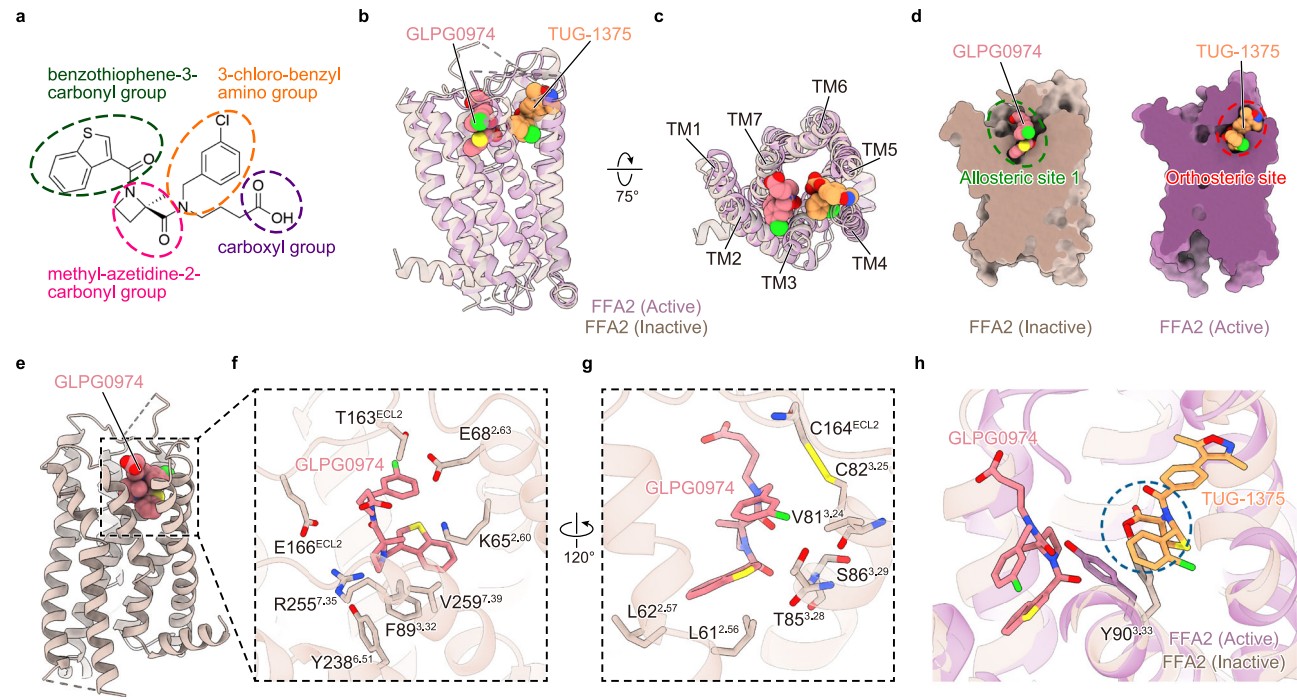

**Fig. 4 | Recognition of GLPG0974. a** Chemical structure of GLPG0974. **b**, **c** Superimposed images of active FFA2 bound to TUG-1375 (purple, orange) and inactive FFA2 bound to GLPG0974 (beige, pink). **d** Cross-section representation of inactive FFA2 bound to GLPG0974 (left) and active FFA2 bound to TUG-1375 (right). **e**–**g** Overall structure of FFA2 bound to GLPG0974 (**e**), and enlarged views of the ligand pocket focused on GLPG0974 (**f**, **g**). **h** Superimposed image of active FFA2 bound to TUG-1375 (purple, orange) and inactive FFA2 bound to GLPG0974 (beige, pink) focused on the ligands. The blue dashed circle indicates a steric clash site between TUG-1375 and inactive FFA2.

GLPG0974 is accommodated within the extracellular binding site, but surprisingly, there is no overlap with the TUG-1375 binding site (Fig. 4b–d). While the binding site of TUG-1375 is mainly formed by TMs 3-5, the binding site of GLPG0974 is mainly formed by TMs 2, 3, and 7 (Fig. 4c). Thus, we term this binding site of GLPG0974 as allosteric site 1.

In allosteric site 1, GLPG0974 exhibits hydrophobic or van der Waals interactions with L61$^{2.56}$, L62$^{2.57}$, V81$^{3.24}$, C82$^{3.25}$, T85$^{3.28}$, S86$^{3.29}$, F89$^{3.32}$, T163$^{ECL2}$, C164$^{ECL2}$, E166$^{ECL2}$, Y238$^{6.51}$, R255$^{7.35}$, and V259$^{7.39}$, a cation-π interaction with K65$^{2.60}$ and an anion-π interaction with E68$^{2.63}$ (Fig. 4e–g). Interestingly, K65$^{2.60}$ has been shown to play a critical role in the species selectivity of GLPG0974[27]. While GLPG0974 inhibits human FFA2, it does not inhibit the phylogenetically close mouse FFA2. This species selectivity has been attributed to the presence of a lysine residue in human FFA2, as previous research has demonstrated that swapping this lysine with the arginine found in mouse FFA2 completely compromises the binding ability of GLPG0974 to human FFA2[27]. Our structure provides a possible explanation for this species selectivity, suggesting that the size difference between Lys and Arg contributes to the selectivity, as the K-to-R substitution in this pocket would cause a steric clash (Fig. 4f).

In addition to species selectivity, GLPG0974 also exhibits subtype selectivity. It specifically inhibits human FFA2 among human FFA receptors and does not inhibit the phylogenetically close human FFA3[27]. Notably, human FFA3 also has an arginine residue at the same position as mouse FFA2 (Supplementary Fig. 5k). This suggests that GLPG0974 likely employs the same mechanism for both species and subtype selectivity, relying on the presence of a lysine residue at position 65$^{2.60}$ in human FFA2.

Another notable feature is observed in the environment surrounding the carboxyl group of GLPG0974. To our knowledge, all reported orthosteric agonists, whether endogenous or synthetic, have a carboxyl group at one end[8,26]. This carboxyl group is assumed to form a salt bridge with the functionally important arginine residue, R255$^{7.35}$, an idea supported by our structural study of TUG-1375 (Fig. 2c). GLPG0974 also shares the carboxyl group, so a previous docking study predicted direct interactions between the carboxyl group and the arginine residue[27]. However, our GLPG0974-bound structure reveals that the carboxyl group does not form a strong direct interaction with any residue in the receptor (Fig. 4f, g). While our MD simulations show a transient interaction with R255$^{7.35}$ via water molecules (Supplementary Fig. 5l), this interaction should be significantly weaker than a direct salt bridge. This explains the results in the previous study showing that the mutation to R255$^{7.35}$ only modestly affects the GLPG0974 binding[16] and that the attachment of a fluorophore to the carboxyl group of GLPG0974 does not affect the binding of the GLPG0974-based fluorescent tracer[28]. These findings suggest that, unlike the role of the carboxyl group in orthosteric agonists, the carboxyl group of GLPG0974 is not a major contributor to its binding and activity.

The unique binding manner of GLPG0974 also suggests an unexpected mechanism of action for this antagonist. As mentioned above, the binding site of GLPG0974 does not overlap with that of TUG-1375 but is positioned just next to it (Fig. 4b–d). Moreover, GLPG0974 binding induces a conformational change in Y90$^{3.33}$, the tyrosine residue dividing the two binding sites, causing it to be pushed towards the TUG-1375 binding site (Fig. 4h). In the TUG-1375-bound structure, Y90$^{3.33}$ actively participates in the recognition of the TUG-1375 agonist (Fig. 2c). However, in the GLPG0974-bound structure, due to the conformational change induced by GLPG0974, Y90$^{3.33}$ collapses the space for the binding of the carboxyl group of TUG-1375, presumably restricting the accessibility of both synthetic and endogenous agonists (Fig. 4h and Supplementary Fig. 5m). These results explain why GLPG0974 can inhibit the binding of orthosteric agonists[16] despite the lack of overlap between their binding sites. Moreover, these

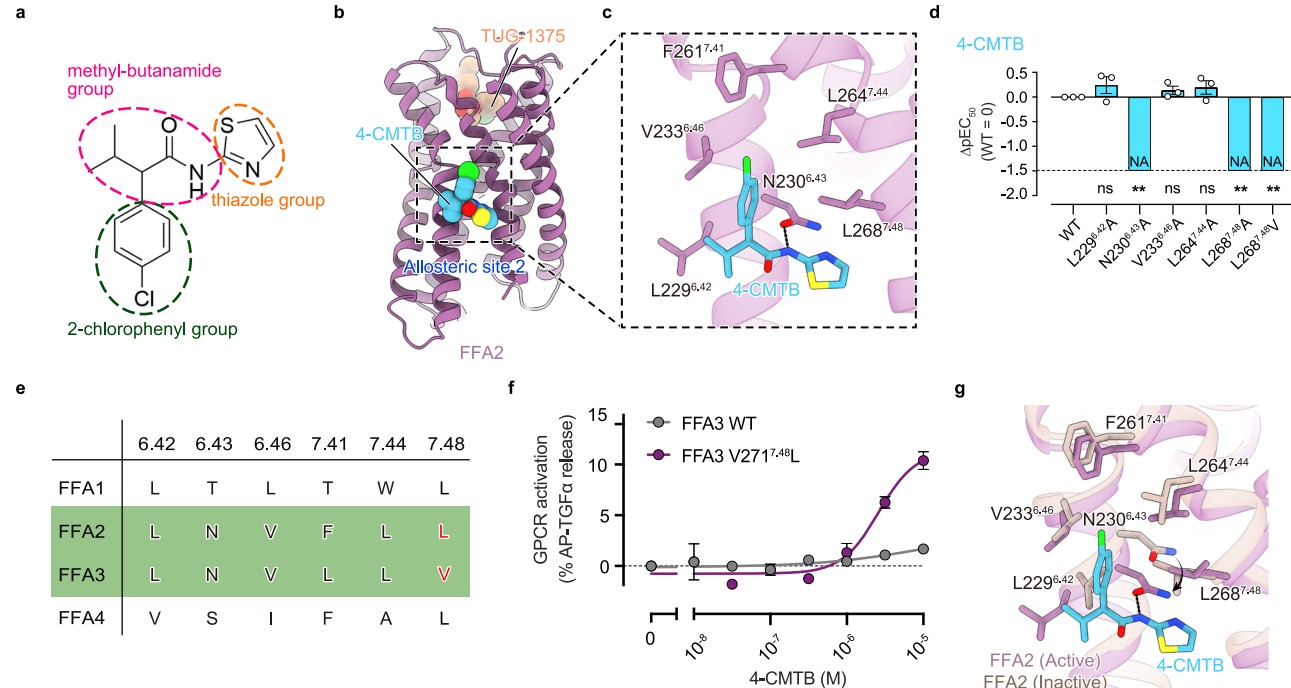

**Fig. 5 | Recognition of 4-CMTB. a** Chemical structure of 4-CMTB. **b, c** Overall structure of FFA2 bound to TUG-1375 and 4-CMTB (**b**), and an enlarged view of the allosteric ligand pocket focused on 4-CMTB (**c**). **d** Mutagenesis analysis of the 4-CMTB-binding site using the TGFα shedding assay. The effect on the potency of 4-CMTB was evaluated by $pEC_{50}$ values. The $pEC_{50}$ values were normalized by that of WT with similar expression levels ($\Delta pEC_{50}$ values) (See also Supplementary Fig. 6a). NA represents parameters not available owing to a lack of ligand response. Bars and error bars represent the mean and SEM, respectively, of three independent experiments, each performed in duplicate. ** represents $p < 0.01$ with one-way ANOVA followed by Dunnett's test for multiple comparison analysis with reference to WT. ns, not significantly different between the groups. See the Source Data file

for additional statistics and exact *p*-values. **e** Sequence comparison of 4-CMTB-binding sites in FFA1-FFA4. **f** Mutagenesis analysis of the FFA3 mutant with an allosteric site mimicking that of FFA2 (V271[7.48]L, purple) using the TGFα shedding assay upon stimulation with 4-CMTB. Symbols and error bars represent the mean and SEM, respectively, of four independent experiments, each performed in duplicate. Note that the symbols for V271[7.48]L at $10^{-7}$ and $10^{-8}$ M are not visible, as the values closely match those of WT. **g** Superimposed image of active FFA2 (purple) and inactive FFA2 (beige) focused on the 4-CMTB-binding site. The black arrow indicates the conformational change of the side chain between inactive and active FFA2.

findings also suggest that, while GLPG0974 has been widely recognized as an orthosteric antagonist[8,26], it should technically be classified as an allosteric antagonist. This is because GLPG0974 does not directly occupy the orthosteric pocket but instead modulates the shape of the orthosteric pocket by shifting the conformation of Y90[3.33], acting as a micro lever switch between the orthosteric and allosteric binding sites.

**Stabilization of active TM 6 and 7 conformations by the allosteric agonist 4-CMTB**

Finally, we focused on 4-CMTB, the best-characterized allosteric agonist of FFA2 (Fig. 5a). 4-CMTB, the first synthetic activator for FFA2[11], has been widely used to analyze the physiology and pathophysiology of FFA2. This compound works as a potent agonist for FFA2 and, at the same time, does not compete with other orthosteric agonists but enhances their potencies, thus classifying it as an allosteric agonist/PAM. Interestingly, the PAM activity of 4-CMTB is ligand-dependent; 4-CMTB acts as a PAM for SCFAs and increases their potencies but does not modulate the potency of other synthetic orthosteric agonists, such as compound 1[29]. This pharmacological characteristic of PAM is called probe dependence[30]. However, not only the mechanisms underlying probe dependence but also the working mechanisms for agonist and PAM activities were elusive. Moreover, despite several previous mutational and computational studies, even the precise binding site of 4-CMTB remained controversial[11,12,31,32].

The cryo-EM map of the active FFA2–Gi complex reveals the binding site of 4-CMTB (Figs. 1a, 5b). Unexpectedly, it does not bind to any previously proposed binding site at the extracellular side of the receptor[11,12,31,32]. Instead, 4-CMTB binds to the middle of the outer

surface of TMs 6 and 7 (Fig. 5b). The entire molecule is surrounded by five hydrophobic residues (L229[6.42], V233[6.46], F261[7.41], L264[7.44], and L268[7.48]), with its methyl-butanamide group forming a hydrogen bond with N230[6.43] (Fig. 5c). We therefore term this binding site as allosteric site 2, and to evaluate the functional significance of these interactions, we generated five single-alanine-substituted mutants and measured their signaling activity upon 4-CMTB stimulation (Fig. 5d and Supplementary Fig. 6a–c). We found that while the L229[6.42]A, V233[6.46]A, and L264[7.44]A mutants showed comparable activity to WT, the L268[7.48]A and N230[6.43]A mutants exhibited over a 30-fold decrease in activity, indicating that the interactions with these two residues are crucial for 4-CMTB function.

To understand the structural basis for the FFA2 specificity of 4-CMTB[29], next we compared the sequences comprising the 4-CMTB binding pocket in FFA2 with those of the other FFA receptors (Fig. 5e). We found that only two of the six residues are conserved in FFA1 and FFA4, explaining why these receptors cannot recognize 4-CMTB. Interestingly, four of the six residues are conserved in FFA3, motivating us to engineer an FFA3 mutant that can be activated by 4-CMTB. We reasoned that V271[7.48] in FFA3 could be a target residue because L268[7.48] is positioned closer to 4-CMTB than F261[7.41] (Fig. 5c), and our functional analysis shows that the mutation of L268[7.48] in FFA2 significantly affects 4-CMTB function (Fig. 5d). Consequently, we mutated V271[7.48] in FFA3 to leucine and tested whether the resulting FFA3 mutant could be activated by 4-CMTB (Fig. 5f and Supplementary Fig. 6d). Remarkably, unlike WT FFA3, the V271[7.48]L mutant was potently activated by 4-CMTB. Notably, 4-CMTB-induced signaling was completely abolished in the L268[7.48]V mutant of FFA2 (Fig. 5d and

Supplementary Fig. 6a–c), indicating that the amino acid positioned at 7.48 is one of the key structural determinants for the subtype selectivity of 4-CMTB.

In general, GPCR activation involves the rotation and displacement of TMs 6 and 7, suggesting that the 4-CMTB binding site observed here is state-dependent and only appears when the receptor adopts the active conformation. The comparison between our active and inactive structures supports this idea; in the inactive state, the methyl-butanamide and thiazole groups have significant steric clashes with L229[6.42] and L268[7.48], respectively, and the methyl-butanamide group loses the hydrogen bond with N230[6.43] (Fig. 5g). The fact that 4-CMTB can only bind to TMs 6 and 7 of active FFA2 indicates that 4-CMTB specifically binds to and stabilizes TMs 6 and 7 in an active conformation. In class A GPCRs, it is widely recognized that there is an equilibrium between active and inactive states for the receptor, even when the receptor binds to orthosteric agonists[33]. Therefore, we propose that 4-CMTB shifts the equilibrium of FFA2 to the active state by binding and stabilizing active TMs 6 and 7. This working model explains the mechanisms not only of how 4-CMTB functions as an allosteric agonist/PAM but also of why 4-CMTB exhibits probe dependence. If the potency of the orthosteric agonist is low, even in the presence of the agonist, the receptor is in equilibrium between active and inactive populations. Thus, 4-CMTB can shift it to the active state, thereby working as a PAM. However, when the potency of the orthosteric agonist is already very high, the binding of the orthosteric agonist alone sufficiently shifts the equilibrium of the receptor to the active population, leaving no room for further shifting by 4-CMTB. This idea is consistent with the fact that SCFAs have much lower potency compared to synthetic orthosteric agonists on which 4-CMTB cannot exert a PAM effect[29], and with several recent studies showing that agonist potency is positively correlated with the population of active states in the equilibrium of the receptor[34,35].

**Proposed FFA2 activation/inactivation mechanisms by orthosteric agonists, allosteric agonists, and allosteric antagonists**

Our FFA2 structures provide an opportunity to analyze the detailed receptor activation and inactivation processes induced by orthosteric agonists, allosteric agonists, and allosteric antagonists (Fig. 6). For activation by orthosteric agonists, whether endogenous or synthetic, the first event would be ligand entry from the extracellular side (Fig. 6a) and recognition of the carboxyl group by polar residues, including Y90[3.33], R180[5.39], Y238[6.51], H242[6.55], and R255[7.35] (Fig. 6b). The binding of the carboxyl group induces a shift of TM6 and TM7 towards the intracellular side (Fig. 6c, d), and this conformational shift is stabilized by several newly formed interactions. In the inactive state, Y94[3.37] forms a hydrogen bond with N239[6.52] (Supplementary Fig. 7a), but the shift of TM6 would switch the hydrogen bonding partner from N239[6.52] to H242[6.55] (Supplementary Fig. 7b). The shift of TM7, together with the conformational shift of Y90[3.33] and subsequent rotation of F89[3.32] (Supplementary Fig. 7c), creates a new hydrophobic interaction between V259[7.36] and F89[3.32], which is further stabilized by the cation-π interaction between K65[2.60] and F89[3.32] (Fig. 6d and Supplementary Fig. 7d, e). The shift of TM7 also affects the Na$^+$ binding site in FFA2. In most class A GPCRs, Na$^+$ binding to D[2.50] and S[3.39] acts as a negative allosteric modulator and stabilizes the receptor in an inactive state[36]. In our inactive structure, four residues (D55[2.50], S96[3.39], N265[7.45], and D269[7.49]) are assembled and form the typical Na$^+$ binding site (Supplementary Fig. 7f). Thus, although we could not identify the Na$^+$ ion due to the limited resolution, it is assumed that Na$^+$ is coordinated at the center of these four residues. The shift of TM7 relocates N265[7.45] and D269[7.49] and collapses the Na$^+$ binding site (Fig. 6e). D269[7.49], a part of the Na$^+$ binding site and the NPxxY motif (DPxxF motif in FFA2), forms a hydrogen bond with N230[6.43] in the inactive state, but this interaction is also broken upon the collapse of the Na$^+$ binding site (Fig. 6e). These interaction losses between TMs 2, 3, 6, and 7 release

TMs 6 and 7 from the receptor core, leading to the rotation and outward shift of their intracellular sides (Fig. 6a, f, g). The displacement of TM7 rearranges the packing at the intracellular side (Supplementary Fig. 7g, h). In the inactive state, F273[7.53] packs against F32[1.57], L51[2.46], L100[3.43], and L272[7.52] (Supplementary Fig. 7g), but after the conformational change of TM7, it packs against L100[3.43] and would stabilize the active conformation (Supplementary Fig. 7h). Notably, while TM7 moves inward upon activation in most class A GPCRs, it moves outward in FFA2 (Fig. 6a). Since the TM7 in the inactive conformation has significant steric clash with the α5 helix of the Gα protein (Supplementary Fig. 7i), not only the outward movement of TM6 but also the outward movement of TM7 is a prerequisite for accommodating the G protein for its activation in FFA2 (Fig. 6h).

In contrast to orthosteric agonists, the allosteric antagonist GLPG0974 also enters the binding pocket from the extracellular side but binds to a site adjacent to the orthosteric site (allosteric site 1) (Figs. 4b, 6i). The antagonist forms a cation-π interaction with K65[2.60], which prevents the formation of the cation-π interaction between K65[2.60] and F89[3.32] that stabilizes the active conformation (Fig. 4f and Supplementary Fig. 7e). In addition, GLPG0974 binding induces a shift of Y90[3.33] in the opposite direction compared to the conformational change caused by orthosteric agonists, collapsing the orthosteric binding pocket (Figs. 4h, 6i).

The allosteric agonist 4-CMTB, on the other hand, accesses the binding pocket not directly from the extracellular bulk solvent but laterally from the lipid bilayer. GPCRs are inherently flexible and exist in an equilibrium between active and inactive states, whether in the apo form or agonist-bound form. During the transition from the inactive to the active state, the intracellular side of TMs 6 and 7 moves outward and rotates clockwise and anticlockwise, respectively (Fig. 6f, g, j). These movements create the binding pocket for 4-CMTB at the outer surface of TMs 6 and 7 (Figs. 5b, 6j). The binding of 4-CMTB to this pocket (allosteric site 2) stabilizes the active conformation of TMs 6 and 7 and shifts the equilibrium of the receptor to the active state. Notably, in the inactive state, N230[6.43] forms a hydrogen bond with D269[7.49] and stabilizes the receptor's inactive conformation (Fig. 6e and Supplementary Fig. 7j). However, the shift and rotation of TMs 6 and 7 upon receptor activation disrupt this interaction, exposing N230[6.43] to the lipid bilayer. This exposure creates a crucial binding site for 4-CMTB. Thus, the binding of 4-CMTB stabilizes the receptor's active conformation, possibly also by indirectly influencing the dynamics of both the Na$^+$ binding pocket and the DPxxF motif through N230[6.43] and D269[7.49], since D269[7.49] is part of both the Na$^+$ binding pocket and the DPxxF motif.

## Discussion

Our TUG-1375–bound FFA2 structure, combined with structural comparisons with FFA1 and gain-of-function mutant experiments, provides insights into the relationships between binding pocket shape, lipid-binding mode, pore opening location, and lipid chain length preference (Fig. 3). In receptors with shallow and narrow binding pockets, the ligand's polar head group inserts deeply into the pocket bottom. Furthermore, pocket orientation correlates with lipid preference: pockets extending to the extracellular bulk solvent (like FFA2) favor SCFAs, while those extending to the lipid bilayer (like FFA1) prefer LCFAs. This pattern appears widespread among human lipid GPCRs. Among 30 reported structures, 12 receptors (FFA2, FFA1, FFA3, GPR34, P2RY10, GPR174, PAFR, GPR183, GPR55, DP2, GPR132, and MRGPRX4) possess shallow binding pockets with basic residues (Arg, Lys, or His) that recognize polar head groups of lipid ligands (Supplementary Fig. 8a). We term these the polar-in group, as their polar head groups consistently insert deeply into the pocket. Within this polar-in group, binding pocket architecture correlates with ligand specificity. For instance, FFA3 specifically recognizes SCFAs through its shallow, narrow pocket extending toward the extracellular side. Conversely,

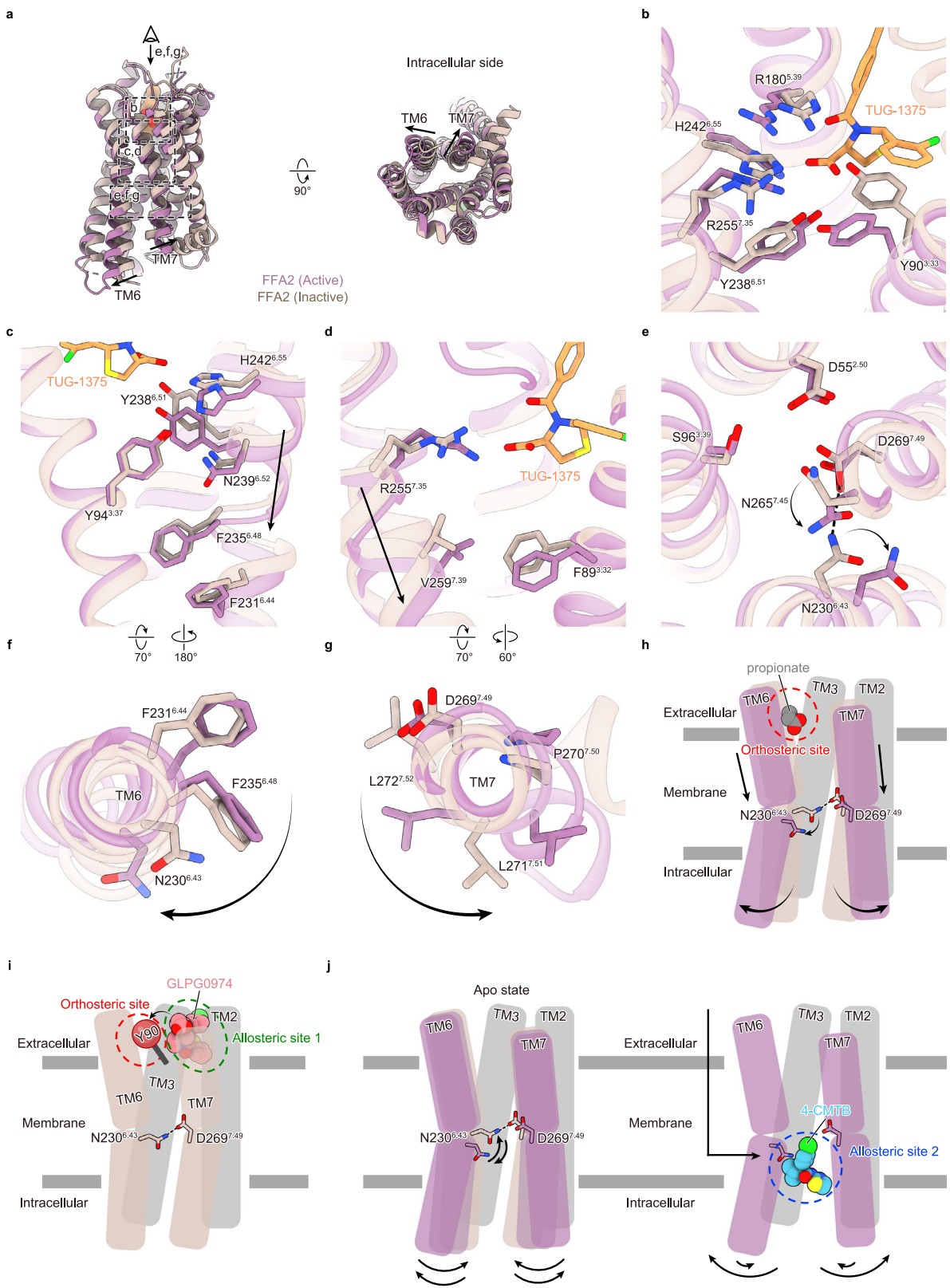

GPR34, P2RY10, GPR174, PAFR, GPR183, and GPR55 have pockets extending toward the lipid bilayer and recognize ligands with long aliphatic moieties. DP2 and GPR132 show more complex architectures, featuring ligand entry sites at the lipid bilayer–solvent boundary or multiple entry points, while still adhering to patterns observed in FFA1 and FFA2. MRGPRX4 appears to be an exception, with its extracellular-facing entry site accommodating large ligands. However, this is not

very surprising because MRGPRX4's ligands with cholesterol moieties (such as DCA-3P and bile acids) are relatively short and compact, and its wider binding pocket allows full accommodation of the ligands without a hydrophobic–hydrophilic mismatch between the ligands and the bulk solvent, unlike the pair of FFA2 and LCFAs.

Notably, in the polar-in group, the shallow pocket positions the lipid head far from the toggle switch—a relatively conserved residue at

**Fig. 6 | Activation and inactivation mechanisms of FFA2. a** Superimposed images of active (purple) and inactive (beige) FFA2. The dashed squares indicate the regions of the views in (**b**–**g**). The eye and arrow symbols indicate the angle of the views in (**e**–**g**). The black arrows indicate the conformational changes of the TM helices between the active and inactive states of FFA2. **b**–**e** Enlarged views of the active and inactive states of FFA2. **b** The rearrangement of the orthosteric binding site. **c**, **d** The shift of extracellular sides of TM6 (**c**) and TM7 (**d**). **e** Collapse of the sodium pocket. Black arrows indicate the downward shift of the transmembrane helices or conformational changes of the side chains. Hydrogen bonds are represented as dashed black lines. **f**, **g** Top views of the active and inactive states of TM6

(**f**) and TM7 (**g**). The black arrows indicate the rotation of the transmembrane helices. **h** A model of FFA2 activation induced by orthosteric agonists. The black arrows indicate the conformational change of the helices and key residue (N230$^{6.43}$) between the inactive and active FFA2. **i** A model of FFA2 inactivation induced by the allosteric antagonist, GLPG0974. The red and green dashed circles indicate the orthosteric site and the allosteric site 1, respectively. **j** A model of FFA2 activation induced by the allosteric agonist, 4-CMTB. The straight black arrow and curved black arrows indicate the ligand access pathway and the flexibility in TM6 and TM7 of FFA2, respectively. The blue dashed circle indicates the allosteric site 2.

position 6.48 proposed to contribute to activation in class A GPCRs—preventing direct interaction between the ligand and the toggle switch involved in receptor activation. Therefore, as observed in FFA2 (Fig. 6a–h), receptors in this group would be activated not through direct ligand–toggle switch interaction but through more complex sequential conformational changes.

Intriguingly, our comparative analysis reveals that 18 of the 30 receptors possess significantly deeper binding pockets compared to those in the polar-in group (Supplementary Fig. 8b). In the polar-out group, the hydrophobic carbon chain, rather than the hydrophilic head, binds deep within the pocket (Supplementary Fig. 8b), and the average distance between the ligand and the toggle switch is consistently shorter (4.6 ± 1.6 Å). Moreover, several studies[37,38] suggest that GPCRs in this group, such as S1P1 and FFA4, are activated through direct interactions between the ligand and the toggle switch. Thus, the ligand recognition and receptor activation mechanisms may be significantly different between the polar-in and polar-out groups. Overall, while further studies are clearly needed, our analysis highlights the key relationships between ligand entry, ligand recognition, and receptor activation mechanisms in lipid GPCRs.

As shown in Fig. 6, our active and inactive FFA2 structures reveal the detailed conformational changes during the receptor activation process. To better contextualize the FFA2 activation mechanism, we compared its structures with three related Gi-coupled GPCRs in both their active and inactive states: μ-opioid receptor (μOR), cannabinoid receptor 1 (CB1), and GPR34. On the extracellular side, FFA2's conformations more closely resemble GPR34 than CB1 and μOR in both active and inactive states (Supplementary Fig. 9a–f). For example, in the active state, while TMs 1, 6, and 7 of FFA2 move inward compared to those of μOR and CB1 (Supplementary Fig. 9a, b, top right), these helices align well with those of GPR34—a finding consistent with FFA2's closer phylogenetic relationship to GPR34 (Supplementary Fig. 9c, top right). On the intracellular side, while the four GPCR structures align well in their active states, their inactive states show notable differences (Supplementary Fig. 9d–f). In FFA2, the intracellular side of TM6 adopts an outward position that appears primed for accommodation of the G protein, unlike μOR, CB1, and GPR34 (Supplementary Fig. 9d–f, bottom right). This suggests that TM7's unique position in the inactive state may play a significant role in preventing G protein binding and subsequent activation.

Notably, recently developed protein prediction software such as AlphaFold2 and AlphaFold-Multistate (AF-Multistate)[39,40] could not accurately predict FFA2's unique activation process, including the outward movement of TM7. The AF-Multistate models, available from the GPCRdb database[41], show consistent active-like conformations in the cytoplasmic regions of TMs 5 and 6 (Supplementary Fig. 10a), while their extracellular sides are markedly different from both active and inactive states (Supplementary Fig. 10b, c). Moreover, these models fail to capture TM7's unique features in FFA2, where it positions closer to the receptor core in the inactive state and moves outward upon activation (Fig. 6a). Instead, in the AF-Multistate models, TM7's position resembles other class A GPCRs, moving inward upon activation (Supplementary Fig. 10a). These findings indicate that while

AlphaFold2 and AF-Multistate are excellent programs for structure prediction, experimentally determined structures, particularly of the inactive state, are essential to reveal the precise activation pathway of FFA2.

In Class A GPCRs, water molecules are reported to influence various aspects of receptor function, including ligand binding, receptor activation, and G-protein coupling[42–44]. Our active and inactive structures, combined with MD simulations, provide a unique opportunity to analyze water distribution in FFA2. Thus, we analyzed water clusters that stably bind within the receptor cavities in each state. As shown in Supplementary Fig. 11, we identified six distinct water clusters located in the receptor cavity. Five clusters appear in both active and inactive states (Supplementary Fig. 11a–e), while one cluster near residues N25$^{1.50}$, T52$^{2.47}$, and D55$^{2.50}$ appears unique to the active state (Supplementary Fig. 11f). Intriguingly, this site typically remains a conserved water-binding site in both active and inactive states across Class A GPCRs[45]. However, in FFA2, the unique position of TM7 in the inactive state causes F273$^{7.53}$ to occupy this site, which becomes available for water molecules only after receptor activation and the accompanying conformational change in F273$^{7.53}$. Thus, unlike other Class A GPCRs, this water cluster may specifically stabilize FFA2's active state and facilitate G-protein binding.

The binding site and working mechanism of GLPG0974 suggest several directions for future study. Although GLPG0974 is a clinical candidate antagonist, further development is clearly needed before it can be used for the treatment of human diseases[17]. One of the reasons why the improvement of GLPG0974 has lagged is that GLPG0974 cannot inhibit FFA2 orthologs of common model animals such as mice and rats[27]. Our current study suggests that the steric clash between the benzothiophene-3-carbonyl group and R$^{2.60}$ in mouse or rat FFA2 is one of the possible reasons for the species specificity of GLPG0974. Since a previous study showed that the benzothiophene-3-carbonyl group can be substituted with other chemical groups[15], further modification of GLPG0974, such as replacing the benzothiophene-3-carbonyl group with a smaller one, may overcome the issue of species specificity and accelerate in vivo studies of FFA2.

Intriguingly, while allosteric site 1 is empty in our cryo-EM structure of active FFA2, our MD simulations of propionate-bound FFA2 suggests that propionate can transiently bind to allosteric site 1 (Supplementary Fig. 4a, 12a, 12b). In one of the 12 simulations, propionate moves between the orthosteric site and allosteric site 1, suggesting that, for endogenous agonists, allosteric site 1 may be important for affinity, efficacy, or both. Designing compounds that target allosteric site 1 could lead to the development of novel modulators that can finely regulate endogenous agonist efficacy, affinity, or both.

Allosteric site 2 may have even higher potential for future drug candidate design. The large displacement of the cytoplasmic sides of TMs 6 and 7 upon receptor activation is commonly observed in class A GPCRs[46]. Thus, it is attractive to hypothesize that every GPCR has a state-dependent allosteric site 2 at the outer surface of TMs 6 and 7. Across druggable class A GPCRs, equivalent residues comprising allosteric site 2 exhibit low sequence homology (approximately 20-

30% identity[47]), suggesting that compounds targeting allosteric site 2 can show high receptor specificity. MIPS521, an allosteric modulator/agonist for the adenosine A1 receptor (A1R), was recently reported to bind to a site close to the 4-CMTB binding site[48] (Supplementary Fig. 13a, b), supporting the potential of allosteric site 2 for the design of receptor- and state-specific allosteric agonists/PAMs for a broad range of class A GPCRs. Notably, the structural comparison of active A1R and FFA2 with and without their allosteric agonists revealed that the binding of 4-CMTB or MIPS521 does not cause further significant conformational changes in allosteric site 2, including the orientation of side chains comprising the site (Supplementary Fig. 13c, d). This implies that the structural information of active GPCRs is sufficient to design the allosteric agonists/PAMs targeting site 2.

Recently, allosteric ligands have gained attention as promising regulators of GPCR function with potential therapeutic benefits[49]. Beyond 4-CMTB and MIPS521, several other allosteric modulators have been reported to target the outer surfaces of TM regions to regulate GPCR activities[50,51]. For example, AS408, an allosteric antagonist/NAM of β2AR, binds to the external surfaces of TMs 3 and 5, inhibiting receptor activity by preventing structural changes in the PIF motif[50] (Supplementary Fig. 13e). In contrast, ZCZ011, an allosteric agonist/PAM of CB1, binds to TMs 2,3, and 4, activating the receptor by destabilizing the Na$^+$ binding pocket[51] (Supplementary Fig. 13f). In addition, our study (Supplementary Fig. 7j) suggests that 4-CMTB binds to TMs 5 and 6, acting as an agonist/PAM by affecting both the Na$^+$ binding pocket and the dynamics of the NPxxY/DPxxF motif. While our understanding of allosteric ligands' diverse binding modes and mechanisms has grown, further research will be essential for comprehensively understanding their working mechanisms and rationally designing next-generation allosteric modulators.

During the preparation of this manuscript, active structures of acetate-, TUG-1375-, and butyrate-bound FFA2 in complex with G proteins were published[52–54] (Supplementary Fig. 13g–j). Structural comparisons indicate that these receptors are highly conserved, particularly within their ligand-binding pockets. The key residues in the orthosteric binding pocket that interact with TUG-1375—Y90$^{3.33}$, R180$^{5.39}$, Y238$^{6.51}$, H242$^{6.55}$, and R255$^{7.35}$—exhibit nearly identical conformations. This supports our proposed mechanism of probe dependence by 4-CMTB: rather than directly affecting the orthosteric binding site, 4-CMTB functions as a PAM by shifting the receptor's equilibrium toward its active state. Furthermore, despite minor structural deviations, the comparison reveals that the carboxyl groups of each ligand are similarly positioned, which is consistent with our MD simulations and mutational analyses (Fig. 2 and Supplementary Fig. 4b). Importantly, our study elucidates the mechanisms underlying fatty-acid chain length preferences, the precise receptor activation processes initiated by orthosteric agonists, and the binding modes and mechanisms of action of the synthetic allosteric antagonist GLPG0974 and the allosteric agonist 4-CMTB. Hence, these previous and current studies are complementary, offering a comprehensive view of FFA2 pharmacology and signaling, which will accelerate the design and development of new drugs for inflammatory and metabolic diseases.

## Methods

### Expression and purification of FFA2

The wild-type human FFA2 (FFA2, UniProtKB O15552; residues 1–325), truncated at the C-terminus, was engineered with an N-terminal Flag-tag and a C-terminal fusion of 2× maltose-binding protein (MBP), monomeric enhanced green fluorescent protein (mEGFP) and a 10× histidine (His) tag. Both the N-terminal and C-terminal tags were designed for removal via cleavage by tobacco etch virus (TEV) protease and human rhinovirus 3 C protease, respectively (See also Supplementary Fig. 1a). ΔGnTI HEK293S cells were cultured in suspension using FreeStyle™293 expression medium (Thermo Fisher Scientific)

until they reached a density of $3.5 \times 10^6$ cells/mL. The cells were then infected with FFA2 baculovirus (Expression Systems) and shaken at 37 °C for 24 hours. 10 mM sodium butyrate (Sigma-Aldrich) was added to the culture and shaken at 30 °C for an additional 24 h. The cell pellets were washed by a low salt buffer (20 mM HEPES−NaOH pH 7.5, 20 mM NaCl, 10 mM MgCl$_2$, 1 mM benzamidine, and 1 µg/mL leupeptin) twice, and were disrupted in a hypertonic lysis buffer (20 mM HEPES−NaOH pH 7.5, 1 M NaCl, 10 mM MgCl$_2$, 1 mM benzamidine, and 1 µg/mL leupeptin) by homogenizing with a glass dounce homogenizer, and the crude membrane fraction was collected by ultracentrifugation (Type 45 Ti rotor, 40,000 rpm ($185,717 \times g$), 1 hour, 4 °C). This process was repeated twice. The membrane fraction was disrupted by homogenizing with a glass dounce homogenizer in a membrane storage buffer (20 mM HEPES−NaOH pH 7.5, 500 mM NaCl, 10 mM imidazole, 20% glycerol, 1 mM benzamidine, and 1 µg/mL leupeptin) and stored at − 80 °C. To purify FFA2 proteins, the membrane fraction was lysed with a solubilization buffer (1% lauryl maltose neopentyl glycol (LMNG, Anatrace), 0.1% cholesteryl hemisuccinate tris salt (CHS), 20 mM HEPES−NaOH pH 7.5, 300 mM NaCl, 5 mM imidazole, 1 mM benzamidine, and 1 µg/mL leupeptin) containing 10 µM TUG-1375 (Axon medchem), 10 µM 4-CMTB (FujiFilm Wako Pure Chemical), and solubilized for 1.5 h at 4 °C. The insoluble cell debris was removed by ultracentrifugation (Type 45 Ti rotor, 40,000 rpm ($185,717 \times g$), 35 min, 4 °C), and the supernatant was mixed with the Ni-NTA superflow resin (QIAGEN) for 1.5 h at 4 °C. The Ni-NTA resin was collected into a glass chromatography column, washed with 18 CV wash buffer (0.01% LMNG, 0.001% CHS, 20 mM HEPES−NaOH pH 7.5, 300 mM NaCl, 30 mM imidazole, 1 µM TUG-1375, and 10 µM 4-CMTB), and was eluted with a wash buffer supplemented with 300 mM imidazole. After cleavage of the 2× MBP-mEGFP-10× His tag with 3 C protease (made in-house), the sample was further purified and concentrated by size-exclusion chromatography on a Superdex 200 Increase 10/300 GL column (Cytiva) in a final buffer (20 mM HEPES−NaOH pH 7.5, 100 mM NaCl, 0.005% LMNG, 0.0005% CHS, 1 µM TUG-1375, and 10 µM 4-CMTB). The peak fractions were pooled and concentrated to approximately 1.5 mg/mL.

### Expression and purification of heterotrimeric Gi

Gi1 heterotrimer was expressed and purified as previously described[18]. In brief, *Trichoplusia ni* insect cells (High Five™, Expression Systems) were co-infected with two viruses, one encoding the wild-type human Gα$_{i1}$ subunit and another encoding the wild-type human Gβ$_1$Gγ$_2$ subunits with an 8× His tag inserted at the N terminus of the Gβ$_1$ subunit. Cultures were collected 48 h after infection. Cells were lysed in a hypotonic buffer, and lipid-modified heterotrimeric Gi1 was extracted in a buffer containing 1% sodium cholate. The soluble fraction was purified using Ni-NTA chromatography, and the detergent was exchanged from sodium cholate to n-dodecyl-β-D-maltoside (DDM, Anatrace) on a column. After elution, the protein was dialyzed against a dialysis buffer (20 mM HEPES−NaOH pH 7.5, 100 mM NaCl, 0.05% DDM, 1 mM MgCl$_2$, 100 µM TCEP, and 10 µM GDP) and concentrated to approximately 20 mg/mL.

### Expression and purification of scFv16

The single-chain construct of Fab16 (scFv16) was expressed and purified as previously described[19]. In brief, a C-terminal 6× His-tagged scFv16 was expressed in secreted form from *Trichoplusia ni* insect cells (High Five™, Expression Systems) using the baculovirus method and purified by Ni-NTA (Qiagen) chromatography. The C-terminal 6× His tag of the Ni-NTA eluent was cleaved by the 3 C protease, and the proteins were dialyzed into a buffer containing 20 mM HEPES−NaOH pH 7.5, and 100 mM NaCl. The sample was reloaded onto the Ni-NTA column to remove the cleaved 6× His. The flow-through containing scFv16 was collected, concentrated, and purified through size-exclusion chromatography on a Superdex 200 10/300 GL column

(Cytiva) in a buffer containing 100 mM NaCl and 20 mM HEPES−NaOH pH 7.5. Monomeric fractions were pooled and concentrated to approximately 100 mg/mL.

## Formation and purification of the FFA2−Gi1−scFv16 complex

To exchange detergent from DDM to LMNG, an equal volume of a buffer containing 20 mM HEPES−NaOH pH 7.5, 50 mM NaCl, 0.5% LMNG, 0.05% CHS, 1 mM $MgCl_2$, and 10 μM GDP was added to purified Gi1, and the protein was incubated at 4 °C for 1 h. Purified FFA2 was mixed with a 1.2 molar excess of Gi1 heterotrimer, and the coupling reaction was allowed to proceed overnight at 4 °C. Apyrase (New England Biolabs), and a 1.25 molar excess of scFv16 were added to hydrolyze unbound GDP and to stabilize the complex, respectively. After one more hour of incubation at 4 °C, the complexing mixture was loaded onto M1 anti-Flag immunoaffinity resin (made in-house). The bound complex was first washed in a buffer containing 0.375% LMNG, followed by washes in gradually decreasing LMNG concentrations and increasing glyco-diosgenin (GDN, Anatrace) concentrations. The complex was then eluted in 20 mM HEPES−NaOH pH 7.5, 50 mM NaCl, 0.00375% LMNG, 0.000375% CHS, 0.00125% GDN, 5% glycerol, 5 μM TUG-1375, 5 μM 4-CMTB, 5 mM EDTA, and 200 μg/mL Flag peptide. The FFA2−Gi1−scFv16 complex was purified by size-exclusion chromatography on a Superdex 200 Increase 10/300 GL column (Cytiva) in 20 mM HEPES−NaOH pH 7.5, 50 mM NaCl, 0.00375% LMNG, 0.000375% CHS, 0.00125% GDN, 1 μM TUG-1375, and 10 μM 4-CMTB. The peak fractions were concentrated to approximately 10 mg/mL for electron microscopy studies.

## Expression and purification of BRIL-fused FFA2

The expression construct for FFA2-BRIL was designed with the insertion of cytochrome b562 RIL (BRIL) into intracellular loop 3 of FFA2. FFA2-BRIL was modified to include an N-terminal Flag-tag epitope, and monomerized enhanced green fluorescent protein (mEGFP), and 10 × His tag; the N-terminal and C-terminal tags are removable by TEV protease and 3 C protease, respectively (See also Supplementary Fig. 2a). The expression and membrane purification process for the FFA2-BRIL was carried out in the same manner as for the wild-type FFA2. To purify FFA2-BRIL, the membrane fraction was lysed with solubilization buffer (1% LMNG (Anatrace), 0.1% CHS, 20 mM HEPES−NaOH pH 7.5, 300 mM NaCl, 5 mM imidazole, 10 μM GLPG0974 (Sigma-Aldrich), 1 mM benzamidine, and 1 μg/mL leupeptin) and solubilized for 1.5 h at 4 °C. The insoluble cell debris was removed by centrifugation (Type 45 Ti rotor, 40,000 rpm ($185,717 × g$), 35 min, 4 °C), and the supernatant was mixed with the Ni-NTA super-flow resin (QIAGEN) for 1.5 h at 4 °C. The Ni-NTA resin was collected into a glass chromatography column, washed with 18 CV wash buffer (0.01% LMNG, 0.001% CHS, 20 mM HEPES−NaOH pH 7.5, 300 mM NaCl, 30 mM imidazole, and 1 μM GLPG0974), and was eluted with a wash buffer supplemented with 300 mM imidazole. Following the cleavage of 10× His-tagged mEGFP by the 3 C protease, the sample was loaded onto the Ni-NTA (Qiagen) column to remove the cleaved His-tagged mEGFP. The flow-through containing FFA2-BRIL was collected, concentrated, and purified by size-exclusion chromatography on a Superdex Increase 200 10/300 GL column (Cytiva) in a final buffer (20 mM HEPES−NaOH pH 7.5, 100 mM NaCl, 0.005% LMNG, 0.0005% CHS, and 1 μM GLPG0974). Peak fractions were pooled and concentrated to approximately 10 mg/mL.

## Expression and purification of BAG2

BAG2 light chain (S1-E215) was modified to include a cysteine (C216) conserved in IgG1 and cloned into a pCAGEN vector containing an N-terminal modified Igκ H (mIgκ H) signal sequence. Heavy chain (E1-S229) was modified to include a cysteine (C230) conserved in IgG1 and cloned into a pCAGEN vector containing an N-terminal modified Igκ H

(mIgκ H) signal sequence and a C-terminal 3 C protease cleavage site followed by an 8 × His tag and twin strep tag. These constructs were expressed at heavy chain: light chain = 1: 1.5 plasmid mass ratio using Expi CHO-S system (Thermo Fisher Scientific). The cultured cells were removed by centrifugation ($6000 × g$, 5 min, RT). Tris−HCl pH 8.0, $NiCl_2$, $CaCl_2$, leupeptin, and benzamidine were added to the supernatant until the concentration of them reached 50 mM, 1 mM, 5 mM, 1 μg/mL, and 1 mM, respectively, and quenched chelate agent in medium for 45 min at RT. The resulting precipitates were removed by centrifugation ($6000 × g$, 20 min, RT) and mixed with the Ni-NTA superflow resin (QIAGEN) for 1 h at RT. The Ni-NTA resin was collected into a glass chromatography column, washed with 20 CV wash buffer (500 mM NaCl, 20 mM HEPES−NaOH pH 7.5, 20 mM imidazole), and eluted in elution buffer (100 mM NaCl, 20 mM HEPES−NaOH pH 7.5, 250 mM imidazole). The eluted Fab was purified by gel-filtration chromatography in a final buffer (100 mM NaCl, 20 mM HEPES−NaOH pH 7.5) using HiLoad 16/600 Superdex 200 pg (Cytiva). The peak fractions were pooled, concentrated, flash-frozen in liquid nitrogen, and stored at − 80 °C until further use.

## Formation and purification of the FFA2-BRIL−BAG2 complex

Purified FFA2-BRIL was mixed with a 1.2-fold molar excess of BAG2 and incubated overnight at 4 °C. The FFA2-BRIL−BAG2 complex was purified by size-exclusion chromatography on a Superdex 200 Increase 10/300 GL column (Cytiva) in a buffer containing 20 mM HEPES−NaOH pH 7.5, 100 mM NaCl, 0.005% LMNG, 0.0005% CHS, and 1 μM GLPG0974. Peak fractions were concentrated to approximately 11 mg/mL for electron microscopy studies.

## Cryo-EM grid preparation

Prior to grid preparation, the sample was centrifuged at $20,380 × g$ for 15 min at 4 °C. The grids were glow-discharged with a PIB-10 plasma ion bombarder (Vacuum Device) at approximately 10 mA current with the dial setting of 2 min for both sides. 3 μL of protein solution was applied to freshly glow-discharged R1.2/1.3 Au 300 mesh holey carbon grid (Quantifoil). Samples were vitrified by plunging into liquid ethane cooled by liquid nitrogen with an FEI Vitrobot Mark IV (Thermo Fisher Scientific) at 4 °C with 100% humidity. The blotting force was set to 10. The waiting and blotting time were 10 s and 4 s, respectively.

## Cryo-EM data acquisition and image processing of FFA2−Gi complex

Cryo-EM images were acquired using a Krios G3i microscope (Thermo Fisher Scientific) equipped with a Gatan BioQuantum energy filter and a K3 direct detection camera in the super-resolution mode, operating at an accelerating voltage of 300 kV. The movie dataset was collected in standard mode using EPU software with a nominal defocus range of − 0.8 to − 1.6 μm. The 8758 movies were acquired at a dose rate of 14.242 e⁻/pixel/s, with a pixel size of 0.83 Å and a total dose of 45.579 e⁻/Å².

Data processing was performed using cryoSPARC v4.3.0[55] and RELION-4.0[56]. The collected 8,758 movies were subjected to Patch motion correction and Patch CTF estimation in cryoSPARC. Initial particles were picked from all micrographs using Template picker and Particle picking with Topaz[57], and were extracted using a box size of 256 pixels. After 2D classification and Heterogeneous refinement, 492,610 particles were selected from 4,703,440 particles. Further classification using Ab-initio reconstruction, removal of duplicated particles, 3D classification with a mask focused on the receptor in RELION, and non-uniform refinement[58], enabled us to obtain a 3.19 Å map with 256,705 particles.

## Cryo-EM data acquisition and image processing of FFA2-BRIL

Cryo-EM images were acquired using a Krios G3i microscope (Thermo Fisher Scientific) equipped with a Gatan BioQuantum energy filter and

a K3 direct detection camera in the super-resolution mode, operating at an accelerating voltage of 300 kV. Using EPU software with a nominal defocus range of − 0.8 to −1.6 μm, 7474 movies were collected in standard mode and 10,021 movies were collected in CDS mode. The 7474 movies were acquired at a dose rate of 14.245 e⁻/pixel/s, with a pixel size of 0.83 Å and a total dose of 45.589 e⁻/Å². The 10,021 movies were acquired at a dose rate of 17.342 e⁻/pixel/s, with a pixel size of 0.83 Å and a total dose of 57.9 e⁻/Å².

Data processing was performed using cryoSPARC v4.4.1 and RELION-4.0. The collected 17,495 movies were subjected to Patch motion correction and Patch CTF estimation in cryoSPARC. Initial particles were picked from all micrographs using Template picker and Particle picking with Topaz and were extracted using a box size of 300 pixels. After 2D classification and Heterogeneous refinement, 538,381 particles were selected from 5,581,238 particles. Subsequent Heterogeneous refinement, 3D classification in RELION using a mask focused on the ligand, non-uniform refinement, and Local refinement with a mask focused on the receptor enabled us to obtain a 3.36 Å map with 76,538 particles.

## Model building and refinement

Initial models of FFA2 and FFA2-BRIL were generated by rigid body fitting of their respective predicted models, which were obtained using the locally installed AlphaFold2[39]. For heterotrimeric G proteins, initial models were formed by rigid body fitting of the S1P1–Gi complex[59] (PDB ID: 7TD4). This resulting starting model was then subjected to iterative rounds of manual refinement in Coot[60] and automated refinement using the Servalcat[61,62], respectively. It is worth noting that GLPG0974 was also fitted to the map and refined using the Servalcat, while the overall pose was analyzed by MD simulation. The final model was visually inspected for general fit to the map, and its geometry was further evaluated using Molprobity[63]. The final refinement statistics are summarized in Supplementary Table 1. All molecular graphics figures were prepared with UCSF Chimera[64] and UCSF ChimeraX[65]. To analyze the receptor activation pathway, the inactive and active FFA2 structures were superimposed using the elements displaying the least conformational changes: TM2, TM3, and the intracellular half of TM4.

## Molecular dynamics simulations of propionate-bound FFA2

We used the structure of FFA2 bound to TUG-1375 presented in this paper as a starting point to perform 12 independent simulations of FFA2 with propionate bound, each at least 1 μs in length.

The structure was prepared for simulation using Maestro (version 2022-3; Schrödinger, LLC). We first removed the G protein as well as the stabilizing scFv from the structure. Since TUG-1375 contains a propionate moiety within it, to model propionate, we simply removed all TUG-1375 atoms that did not correspond to the propionate moiety. Missing amino acid side chains were modeled using Prime (Schrödinger, LLC). Neutral acetyl and methylamide groups were added to cap the N- and C-termini, respectively, of the protein chains. All histidine residues were modeled as neutral with a proton on the epsilon nitrogen. Other titratable residues were kept in their dominant protonation state at pH 7.4, except for D55²·⁵⁰, E106³·⁴⁹, E182⁵·⁴¹, and D269⁷·⁴⁹, which were protonated (neutral) due to the side chains being buried within the protein or facing the membrane lipids. Dowser[66] was used to add water molecules to protein cavities. We used the Orientations of Proteins in Membranes (OPM) PPM 3.0 web server[67] to align the proteins, after which the aligned structures were inserted into a pre-equilibrated palmitoyl oleoyl-phosphatidylcholine (POPC) membrane bilayer using Dabble[68] (version 2.7.9). Sodium and chloride ions were added at a concentration of 150 mM to neutralize the system. The final system contained 49,264 atoms, including 125 lipid molecules and 9319 water molecules (initial system dimensions: 80 Å × 77 Å × 84 Å). A summary of the simulation setup is provided in Supplementary Table 2.

Initial atom velocities were assigned randomly and independently for each simulation. We employed the CHARMM36m force field for protein molecules, the CHARMM36 parameter set for lipid molecules and salt ions, and the associated CHARMM TIP3P model for water molecules[69,70]. Simulations were run using the AMBER20 software (AMBER20, University of California, San Francisco) under periodic boundary conditions with the Compute Unified Device Architecture (CUDA) version of Particle-Mesh Ewald Molecular Dynamics (PMEMD)[71] on one GPU.

The systems were first heated over 12.5 ps from 0 K to 100 K in the NVT ensemble using a Langevin thermostat with harmonic restraints of 10.0 kcal·mol⁻¹·Å⁻² on the non-hydrogen atoms of the lipids, ligand, and protein. Initial velocities were sampled from a Maxwell-Boltzmann distribution. The systems were then heated to 310 K over 125 ps in the NPT ensemble. Equilibration was performed at 310 K and 1 bar in the NPT ensemble, with harmonic restraints on the protein and ligand non-hydrogen atoms tapered off by 1.0 kcal·mol⁻¹·Å⁻² starting at 5.0 kcal·mol⁻¹·Å⁻² in a stepwise manner every 2 ns for 10 ns, and finally by 0.1 kcal·mol⁻¹·Å⁻² every 2 ns for an additional 18 ns. All restraints were completely removed during production simulations. Production simulations were performed at 310 K and 1 bar in the NPT ensemble using the Langevin thermostat and Berendsen barostat.

Lengths of bonds to hydrogen atoms were constrained using SHAKE, and the simulations were performed using a time step of 4.0 fs while employing hydrogen mass repartitioning[72]. Non-bonded interactions were cut off at 9.0 Å, and long-range electrostatic interactions were calculated using the particle-mesh Ewald (PME) method with an Ewald coefficient (β) of approximately 0.31 Å⁻¹ and B-spline interpolation of order 4. The PME grid size was chosen such that the width of a grid cell was approximately 1 Å. Snapshots of the trajectory were saved every 200 ps.

The AmberTools17 CPPTRAJ package[73] was used to reimage trajectories at 1 ns per frame. Visual Molecular Dynamics (VMD; version 1.9.4a57)[74] and vmd-python (version 3.0.6) were used for visualization and analysis.

To monitor the dynamics of propionate, we calculated the root-mean-square deviation (RMSD) of non-hydrogen propionate atoms to their initial positions after aligning on all protein backbone atoms.

## Molecular dynamics simulations of GLPG0974-bound FFA2

Hydrogen atoms were generated and energetically optimized with the heavy atom positions fixed using the CHARMM program[75] (version 35b2). Atomic charges and force field parameters of the amino acids were obtained from the CHARMM22 parameter set[76]. Atomic charges of GLPG0974 with deprotonated carboxyl group were determined by fitting the electrostatic potential using the restrained electrostatic potential (RESP) procedure[77]. The electronic wave functions were calculated after geometry optimization. The restricted density functional theory method was employed with the B3LYP functional and LACVP* basis sets, using the JAGUAR program (version 7.9; Schrödinger, LLC). Force field parameters of GLPG0974 were obtained from the generalized Amber force field[78].

The protonation pattern of the protein was determined based on the electrostatic continuum model, solving the linear Poisson-Boltzmann equation with the MEAD program[79] (version 2.2.9). The experimentally measured $pK_a$ values employed as references were 12.0 for Arg, 4.0 for Asp, 9.5 for Cys, 4.4 for Glu, 10.4 for Lys, 9.6 for Tyr[80], and 7.0 and 6.6 for the $N_\varepsilon$ and $N_\delta$ atoms of His, respectively[81–83]. The dielectric constants were set to 4 for the protein interior and 80 for water. All computations were performed at 300 K, pH 7.0, and with an ionic strength of 100 mM. The linear Poisson-Boltzmann equation was solved using a three-step grid-focusing procedure at resolutions of 2.5, 1.0, and 0.3 Å. The ensemble of the protonation pattern was sampled by the Monte Carlo method with the Karlsberg program[84] (version 1.0.2). R219 and R255 were kept protonated in the MD simulations,

although the calculated protonation probabilities were 40% and 28–31%, respectively, as these Arg residues are exposed to the bulk water region. The resulting protonation pattern consists of protonated D55, D269, E68, E166, and E182; deprotonated K79; and standard protonation states for other titratable residues (i.e., deprotonated Asp and Glu, protonated Arg and Lys, and neutral His with a proton on the epsilon nitrogen). The calculated protonation probability of K65$^{2.60}$ was 35% for the pose1-bound structure and 95% for the pose2-bound structure, respectively. Because K65$^{2.60}$ forms the binding pocket of GLPG0974, the protonation state of K65$^{2.60}$ might affect the binding mode of GLPG0974. Therefore, we conducted MD simulations with both deprotonated and protonated K65$^{2.60}$.

The FFA2 assembly was embedded in a lipid bilayer consisting of 269 POPC molecules using CHARMM-GUI[85], and soaked in 23,660–23,662 TIP3P water models. 44 sodium and 60–61 chloride ions were added to neutralize the system with an ionic strength of 100 mM using the VMD plugins[74] (version 1.9.2). After structural optimization with position restraints on heavy atoms of the FFA2 assembly, the system was heated from 0.1 to 300 K over 5.5 ps with a time step of 0.1 fs, equilibrated at 300 K for 1 ns with a time step of 0.5 fs, and annealed from 300 to 0 K over 5.5 ps with a time step of 0.1 fs. The positional restraints on heavy atoms of side chains were released, and the same procedure was repeated. Positional restraints on any atoms were released, and the system was heated from 0.1 K to 300 K over 5.5 ps with a time step of 0.1 fs and equilibrated at 300 K for 1 ns with a time step of 0.5 fs. Finally, a production run was conducted for 500 ns at 300 K and 1 bar with a time step of 1.5 fs. Three independent MD simulations were conducted for each GLPG0974 pose and K65$^{2.60}$ protonation state. All MD simulations were conducted using the MD engine NAMD[86] (version 2.13). For MD simulations with a time step of 1.5 fs, the SHAKE algorithm for hydrogen constraints was employed[87]. For temperature and pressure control, the Langevin thermostat and piston were used[88]. To quantify the conformation of GLPG0974, we calculated the RMSD of non-hydrogen GLPG0974 atoms with respect to the cryo-EM model after aligning on all protein backbone atoms.

### TGFα shedding assay

Ligand-induced FFA receptor activation was measured by the TGFα shedding assay[89], with minor modifications. HEK293A cells (Thermo Fisher) were seeded in a 6 cm culture dish (Greiner Bio-One) at a concentration of $2 \times 10^5$ cells/mL (4 mL per dish hereafter) in DMEM (Nissui Pharmaceutical) supplemented with 5% (v/v) FBS (Gibco), glutamine, penicillin, and streptomycin (complete DMEM), one day before transfection. The transfection solution was prepared by combining 10 μL of 1 mg/mL polyethylenimine Max solution (Polysciences) and a plasmid mixture consisting of 400 ng ssHA-Flag-FFA receptor, 1 μg alkaline phosphatase (AP)-tagged TGFα (AP-TGFα; human codon-optimized), and 600 ng empty pCAGGS plasmid. For evaluation of the ligand-induced response of FFA3, 200 ng of Gαq/i1 was transfected, along with plasmids encoding GPCR and AP-TGFα. Note that the full-length human GPCR constructs were inserted into the pCAGGS expression vector with an N-terminal fusion of the hemagglutinin-derived signal sequence (ssHA), Flag epitope tag, and a flexible linker (MKTIIALSYIFCLVFA<u>DYKDDDDK</u>GGSGGGGSGGSSSGGG; the Flag epitope tag is underlined). One day after incubation, the transfected cells were harvested by trypsinization, neutralized with the complete DMEM, washed once with Hank's Balanced Salt Solution (HBSS) containing 5 mM HEPES–NaOH (pH 7.4), and resuspended in 12 mL of the HEPES-containing HBSS. The cell suspension was seeded into a 96-well plate (Greiner Bio-One) at a volume of 90 μL (per well hereafter) and incubated for 30 min in a CO$_2$ incubator. Test agonists diluted in 0.01% (w/v) BSA- and the HEPES-containing HBSS (at 10 × concentration) or vehicle were added at a volume of 10 μL, and the plate was incubated for 1 h. After centrifugation, conditioned media (80 μL) were transferred to an empty 96-well plate. AP reaction solution (10 mM p-

nitrophenyl phosphate (p-NPP), 120 mM Tris–HCl (pH 9.5), 40 mM NaCl, and 10 mM MgCl$_2$) was dispensed into the cell culture plates and the conditioned media plate at a volume of 80 μL each. Absorbance at 405 nm was measured before and after 1 h incubation at room temperature using a microplate reader (SpectraMax340PC384, Molecular Devices). Unless otherwise noted, vehicle-treated AP-TGFα release signal was set as a baseline. AP-TGFα release signals were fitted by the "Nonlinear Regression: Variable slope (four parameter)" in the Prism 10 tool with a constraint of the absolute Hill Slope values less than 2. For each experiment, the parameters Span (= Top – Bottom) and pEC$_{50}$ of individual receptor mutants were normalized to those of the wild-type receptor performed in parallel, and the resulting $E_{max}$ and pEC$_{50}$ values were used to calculate the ligand response of the mutants.

### Flow cytometry

Cellular surface expression of the GPCR constructs was measured with flow cytometry[90]. Plasmid transfection for the ssHA-Flag-GPCR and the AP-TGFα reporter was performed according to the same procedure as described in the TGFα shedding assay section. One day after transfection, the cells were collected by adding 200 μL of 0.53 mM EDTA-containing Dulbecco's PBS (D-PBS), followed by 200 μL of 5 mM HEPES–NaOH (pH 7.4)-containing HBSS. The cell suspension was transferred to a 96-well V-bottom plate in duplicate and fluorescently labeled with an anti-Flag epitope (DYKDDDDK) tag monoclonal antibody (Clone 1E6, FujiFilm Wako Pure Chemicals, cat. no. 012-22384; 10 μg/mL diluted in 2% goat serum- and 2 mM EDTA-containing D-PBS (blocking buffer)) and a goat anti-mouse IgG secondary antibody conjugated with Alexa Fluor 488 (Thermo Fisher Scientific, cat. no. A11001; 10 μg /mL diluted in the blocking buffer). After washing with D-PBS, the cells were resuspended in 200 μL of 2 mM EDTA-containing D-PBS and filtered through a 40-μm filter. The fluorescent intensity of single cells was quantified by an EC800 flow cytometer equipped with a 488 nm laser (Sony). The fluorescent signal derived from Alexa Fluor 488 was recorded in an FL1 channel, and the flow cytometry data were analyzed with the FlowJo software (FlowJo). Live cells were gated with a forward scatter (FS-Peak-Lin) cutoff at the 390 setting, with a gain value of 1.7. Values of mean fluorescence intensity (MFI) from ~20,000 cells per sample were used for analysis. For each experiment, we normalized an MFI value of the mutants by that of WT performed in parallel and denoted relative levels.

### Statistical analysis

For the functional analysis, statistical analyses were performed using the GraphPad Prism 10 (Ver 10.1.0) software (GraphPad), and the methods are described in the legends of the figures. The representation of symbols and error bars is described in the legends. Symbols are mean values of indicated numbers of independent experiments. Unless otherwise noted, error bars and shaded areas denote SEM. For multiple comparison analysis, one-way ANOVA followed by the Dunnett's test was used.

### Reporting summary

Further information on research design is available in the Nature Portfolio Reporting Summary linked to this article.

## Data availability

The raw images of TUG-1375/4-CMTB-bound FFA2–Gi complex and GLPG0974-bound FFA2-BRIL before motion correction have been deposited in the Electron Microscopy Public Image Archive under accession EMPIAR-12493 [10.6019/EMPIAR-12493]. The cryo-EM density map and atomic coordinates for TUG-1375/4-CMTB-bound FFA2–Gi complex and GLPG0974-bound FFA2-BRIL have been deposited in the Electron Microscopy DataBank: EMD-39003 and EMD-39004, and PDB under accessions: 8Y6W and 8Y6Y, respectively. The

simulation data for propionate-bound FFA2 and GLPG0974-bound FFA2-BRIL have been deposited on Zenodo: entry 14853893 and 14885834, respectively. All other data are provided in this article, its Supplementary Information and Source Data file, or from the corresponding author upon request. Source data are provided in this paper.

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

## Acknowledgements

We thank T. Kusakizako and Y. Sakamaki (UTokyo) for help with cryo-EM data collection, K. Sato, S. Nakano, and A. Inoue (Tohoku Univ.) for their assistance in plasmid preparation and the cell-based GPCR assays, and K. Hasegawa and H. Yasumoto (UTokyo) for administrative support. We also acknowledge Claude 3, a multimodal large language model created by Anthropic, for providing guidance to improve the readability of this manuscript. Note that, after using this tool, we reviewed and edited the content as needed and took full responsibility for the content of the publication. This work was supported by the Interdisciplinary Computational Science Program in CCS, University of Tsukuba (K.S.), JSPS KAKENHI (JP23KJ0363/JP24K18286 to K.Kawakami, JP22KJ1109 to M.T., JP23H04963/JP24K01986 to K.S., JP23H02444 to H.Ishikita, JP21H04791/JP21H05113/JP21H05037/JP24K21281 to A.I. JP21H04791/JP21H05142 to H.E.K.), JST SPRING (JPMJSP2108 to M.K.), JST FOREST (JPMJFR215T to A.I., JPMJFR204S to H.E.K.), JST CREST (JPMJCR21P3/JPMJCR23B1 to H.E.K.), JST moonshot R&D (JPMJMS2023 to A.I.), AMED (JP22ama121038/JP22zf0127007 to A.I., 24bm1123057h0001 to H.E.K.), the National Institutes of Health (NIH) (R01GM127359 to R.O.D.), and the Senri Life Science Foundation (K.Kawakami) and the Uehara Memorial Foundation (K.Kawakami).

## Author contributions

M.K., M.F., T.E.M., and J.K. prepared purified proteins and cryo-EM grids of the FFA2–Gi complex. M.K. and M.F. obtained cryo-EM images of the FFA2–Gi complex. M.K. and T.E.M. processed cryo-EM data of the FFA2–Gi complex. M.K., M.F., and T.E.M. built the model and refined the structure of the FFA2–Gi complex. M.K., K. Kobayashi, A.K., W.J.I., M.F., and S.K. prepared purified proteins and cryo-EM grids of FFA2-BRIL. M.K., K. Kobayashi, A.K., W.J.I., and M.F. obtained cryo-EM images of FFA2-BRIL. M.K., K. Kobayashi, A.K., and W.J.I. processed cryo-EM data of FFA2-BRIL. M.K., K.Kobayashi, and M.F. built the model and refined the structure of FFA2-BRIL with the help of K.Y. R.K., K. Kawakami, and A.F. performed mutagenesis analysis, supervised by A.I. C.-M.S. performed and analyzed MD simulations of FFA2 bound to propionate, with guidance from R.O.D. M.T. performed MD simulations of FFA2 bound to GLPG0974, supervised by K.S. and H. Ishikita. H.Ikeda conducted cloning and mutagenesis. M.K., K. Kawakami, and H.E.K. wrote the manuscript with input from all authors. A.I. and H.E.K. supervised all aspects of the research.

## Competing interests

R.O.D. serves on the scientific advisory board of Septerna. The other authors declare no competing interests.
