## [Transparent Peer Review file · Nature Communications]

Structural insights into lipid chain-length selectivity and allosteric regulation of FFA2

Corresponding Author: Dr Hideaki Kato

Version 0:

Reviewer comments:

Reviewer #1

(Remarks to the Author)

The author used cryo-EM, mutagenesis, and computational modeling to investigate ligand recognition and activation of the FFA2 receptor. The results generally support the manuscript's claims, particularly the identification of two allosteric sites, which is crucial for selective drug discovery. I believe this manuscript is suitable for publication in Nature Communications. However, I have a few suggestions before publication:

1. In Extended Data Figures 1 and 2, please include SDS-PAGE analyses for the complex samples to demonstrate the purity and composition of the protein complexes used in the cryo-EM studies. The authors present two FFA2 structures bound to different types of ligands. I suggest comparing these structures with other related receptor structures (both active and inactive) to better contextualize the conformational changes and clearly delineate the active or inactive states of FFA2.
2. The lipid entry site in FFA2 appears to share similarities with other lipid receptors. Please compare this site with related receptors (e.g., DOI: 10.1038/nature20613) and comment on any conserved features related to ligand entry or recognition.
3. The authors propose that GLPG0974 acts as an allosteric antagonist by occupying a different site than the orthosteric agonist. Did the authors measure the inhibition efficacy of GLPG0974 in receptors stimulated by different agonists? I am particularly interested in the potential probe dependence of GLPG0974 as an allosteric antagonist, similar to the case presented in DOI: 10.1038/s41392-023-01625-y. Please explore this feature further.
4. Given the unexpected mode of action of GLPG0974, I suggest conducting additional experiments to clarify this novel antagonistic behavior. In Extended Data Figure 5, the author validated the ligand-binding poses using MD simulation, I recommend extending the MD simulations to at least 500 ns to ensure a more robust evaluation of ligand-receptor interactions.
5. The manuscript highlights the probe dependence of 4-CMTB, which enhances the potency of SCFAs but not synthetic agonists like TUG-1375. Have the authors tested 4-CMTB's effects with other classes of ligands beyond SCFAs and TUG-1375 to confirm this observation?
6. The binding of 4-CMTB to the outer surface of TM6-TM7 and its stabilization of this conformation is intriguing. Does this PAM effect involve any classical motifs typically associated with receptor activation? For instance, an allosteric modulator of the beta-2 adrenergic receptor binds outside of TM5-TM6 and regulates the P-I-F motif. Please provide a comparative discussion on this point.

Reviewer #2

(Remarks to the Author)

Two sets of MD simulations were performed to propose and refine structural models and describe atomic interactions of the FFA2 receptor in complex with 1) propionate, an orthosteric agonist, and 2) the orthosteric antagonist GLPG0974.

In the propionate-receptor complex system, the propionate stability was analyzed through 12 independent simulations. For the GLPG0974-receptor complex system, the protonated and deprotonated state of K652 was performed to define the stability of the two GLPG0974 binding modes. The simulations were performed in POPC lipid membranes.

The CHARMM36M and CHARMM22 force files were used for proteins, lipids, and ions, which are well-established and

validated for transmembrane proteins. The CHARMM TIP3P model was used for water, which has been thoroughly validated.

The authors performed multiple independent simulations that enhanced their data's reliability and validated the simulations' sampling.

Points that need to be addressed.

* The figures, figure legends, and figure references from the extended data DO NOT CORRESPOND. Therefore, the paper was not easy to follow. It should be corrected!

* It is unclear why different protocols were used to address the protonation states of the simulated systems. The use of only two protonation states for K652.60 in GLPG0974 might oversimplify the actual range of possible states.

* The authors found a key water molecule forming a water bridge between R255 and the GLPG0974 ligand. What happened to the waters through the simulated systems inside the protein cavities? Do the active and inactive conformation systems contain different water arrangements and dynamics in the protein cavities/channel?

Reviewer #3

(Remarks to the Author)

In this study, Mai Kugawa et al., determined the cryo-EM structures of active FFA2-Gi complex with the orthostatic agonist orthosteric agonist TUG-1375 and the PAM 4-CMTB, and the inactive FFA2 structure bound to the antagonist GLPG0974. By comparing these structures, together with MD simulations and functional studies, these findings shed light on the ligand recognition and activation of orthosteric / allosteric ligands. It also uncovers the binding site and mechanisms of clinically relevant antagonist GLPG0974. In general, this is an informative paper with well organised presentation of the data. The cryo-EM maps and preliminary PDB reports reflects the good quality of the structure deposited.

Minor comments as below:

1. Have the constructs of FFA2 used in this study been validated? Do they make pharmacological sense? These data should be included. Or the references should be given if they have been validated in previous studies.
2. Can the PAM 4-CMTB modulate the synthetic orthosteric agonist TUG-1375? If not, why did the authors choose this combination? And will the binding of PAM cause any effect on the orthosteric ligand binding? It is worthwhile to further discuss these in the discussion part.
3. The side chain rotamer of most key residues discussed in this study were well supported by the cryo-EM map density. However, the density of residues 65, 242, 255 of the inactive structure are not so well resolved, and there might be different ways of modelling. Therefore, I am not so convinced of the related activation / inactivation mechanisms presented in Fig. 6.
4. In the extended Fig.7a, why is the agonist TUG-1375 present in an inactive structure?
5. As there are relevant structures published more than half year ago (Ref 43, 44), particularly the FFA2-Gi active structures bound to the same ligand TUG-1375 or endogenous SCFA acetate, which directly related to this study. I would suggest the authors to fully discuss these structures.

Version 1:

Reviewer comments:

Reviewer #1

(Remarks to the Author)

The authors have addressed my concerns.

Reviewer #2

(Remarks to the Author)

I have no further questions. My points have been adequately addressed.

Reviewer #3

(Remarks to the Author)

All my concerns have been addressed in the revisions, and I believe this second version of manuscript is suitable for publication in Nature Communications.

Reviewers' comments:

Reviewer #1

The author used cryo-EM, mutagenesis, and computational modeling to investigate ligand recognition and activation of the FFA2 receptor. The results generally support the manuscript's claims, particularly the identification of two allosteric sites, which is crucial for selective drug discovery. I believe this manuscript is suitable for publication in Nature Communications.

We greatly appreciate the positive comments and support.

However, I have a few suggestions before publication:

1. In Extended Data Figures 1 and 2, please include SDS-PAGE analyses for the complex samples to demonstrate the purity and composition of the protein complexes used in the cryo-EM studies.

We appreciate this suggestion and added the gel images from the SDS-PAGE analyses to **Supplementary Figs. 1c and 2b (= Response Fig. 1).**

Response Fig. 1 (= Supplementary Figs. 1c and 2b) SDS-PAGE analyses.

a,b, SDS-PAGE analyses of the FFA2–G α i–scFv16 complex (**a**) and the FFA2-BRIL–BAG2 complex (**b**). HC and LC indicate heavy chain and light chain of BAG2, respectively.

The authors present two FFA2 structures bound to different types of ligands. I suggest comparing these structures with other related receptor structures (both active and

inactive) to better contextualize the conformational changes and clearly delineate the active or inactive states of FFA2.

We thank Reviewer #1 for this valuable suggestion. Following Reviewer #1's suggestion, we compared the structures of FFA2 with three representative Gi-coupled GPCRs in both their active and inactive states:

- μ OR (a prototypical Gi-coupled GPCR; active: PDB ID 5C1M, inactive: 4DKL)
- CB1 (a Gi-coupled lipid receptor; active: 7WV9, inactive: 5U09)
- GPR34 (a Gi-coupled lipid receptor closely related to FFA2; active: 8SAI, inactive: 8IYX)

First, on the extracellular side, FFA2's conformations more closely resemble to GPR34 than CB1 and μ OR in both active and inactive states (Response Fig. 2a–f). For example, in the active state, while TMs 1, 6, and 7 of FFA2 move inward compared to those of μ OR and CB1 (Response Fig. 2a–b, top right), these helices align well with those of GPR34—a finding consistent with FFA2's closer phylogenetic relationship to GPR34 (Response Fig. 2c, top right). On the intracellular side, while the four GPCR structures align well in their active states, their inactive states show notable differences (Response Fig. 2d–f). In FFA2, the intracellular side of TM6 adopts an outward position that appears primed for accommodation of the G protein, unlike μ OR, CB1, and GPR34 (Response Fig. 2d–f, bottom right). This suggests that TM7's unique position in the inactive state may play a significant role in preventing G protein binding and subsequent activation.

We added Response Fig. 2 as Supplementary Fig. 9 and included the results of this comparison in the discussion section of the main manuscript to clarify the structural differences and similarities between FFA2 and related GPCRs (i.e., μ OR, CB1, and GPR34).

Response Fig. 2 (= Supplementary Fig. 9) Structural comparison of FFA2 and other GPCRs. **a-c**, Superimposed images of the active FFA2 structure with μ OR (**a**, 5C1M), CB1 (**b**, 7WV9), and GPR34 (**c**, 8SAI). **d-f**, Superimposed images of the inactive FFA2 structure with μ OR (**d**, 4DKL), CB1 (**e**, 5U09), and GPR34 (**f**, 8IYX).

2. The lipid entry site in FFA2 appears to share similarities with other lipid receptors. Please compare this site with related receptors (e.g., DOI: 10.1038/nature20613) and comment on any conserved features related to ligand entry or recognition.

Our initial manuscript compared the lipid entry sites among FFA2, FFA1 (Fig. 3), FFA3, GPR174, GPR34, GPR132, and GPR183 (Extended Data Fig. 8). Following the reviewer's suggestion, we expanded our analysis to include all other lipid GPCRs with reported structures, excluding those lacking observable ligand densities or those bound to antagonists. This comprehensive analysis of 30 GPCR structures revealed two distinct classifications: "polar-in" and "polar-out." Additionally, the key relationships among ligand entry site, binding pocket shape, binding mode, and lipid chain length preference observed in FFA1 and FFA2 appear to be conserved across receptors in the "polar-in" group.

First, among the 30 GPCR structures, 12 receptors (FFA2, FFA1, FFA3, GPR34, P2RY10, GPR174, PAFR, GPR183, GPR55, DP2, GPR132, and MRGPRX4) possess shallow binding pockets with basic residues (Arg, Lys, or His) that recognize the polar head groups of lipid ligands (Supplementary Fig. 8a). We term these the "polar-in" group, as their polar head groups consistently insert deeply into the pocket. Within this group, binding pocket architecture correlates with ligand specificity. For instance, FFA3 specifically recognizes SCFAs through its shallow, narrow pocket extending toward the extracellular side. Conversely, GPR34, P2RY10, GPR174, PAFR, GPR183, and GPR55 have pockets extending toward the lipid bilayer and recognize ligands with long aliphatic moieties. DP2 and GPR132 exhibit more complex architectures, featuring ligand entry sites at the lipid bilayer-solvent boundary or multiple entry points, while still adhering to patterns observed in FFA1 and FFA2. MRGPRX4 appears to be an exception, with its extracellular-facing entry site accommodating large ligands. However, this is not surprising because MRGPRX4's ligands, which contain cholesterol moieties (such as DCA-3P and bile acids), are relatively short and compact. Its wider binding pocket allows full accommodation of these ligands without a hydrophobic-hydrophilic mismatch between the ligands and the bulk solvent, unlike the pair of FFA2 and LCFAs.

Notably, in the "polar-in" group, the shallow pocket positions the lipid head far from the "toggle switch"—a relatively conserved residue at position 6.48 proposed to contribute to activation in class A GPCRs—preventing direct interaction between the ligand and the "toggle switch" involved in receptor activation. Therefore, as observed in FFA2 (Fig. 6a–h), receptors in this group likely achieve activation through complex sequential conformational changes rather than direct ligand-toggle switch interactions.

In contrast to receptors in the "polar-in" group, 18 of the 30 receptors have significantly deeper binding pockets (Supplementary Fig. 8b) and comprise the "polar-out" group. In this group, ligand entry sites appear to be diverse. This divergence is reasonable because, even when the ligand entry site opens toward the extracellular bulk solvent and ligands extend to this side, it creates no significant hydrophobic-hydrophilic mismatches since the pocket is deep enough to accommodate the entire ligand. In the "polar-out" group, the hydrophobic carbon chain, rather than the hydrophilic head, consistently localizes deep within the pocket

(Supplementary Fig. 8b), and the average distance between the ligand and the toggle switch is consistently shorter ($4.6 \pm 1.6 \text{ \AA}$). Moreover, several studies^{1,2} suggest that GPCRs in this group, such as S1P1 and FFA4, are activated through direct ligand–toggle switch interactions. Consequently, the ligand recognition and receptor activation mechanisms are significantly different between the “polar-in” and “polar-out” groups.

We added Response Fig. 3 as Supplementary Fig. 8 and included the results of this analysis in the discussion section of the main manuscript.

7 α ,25-dihydroxycholesterol-bound GPR183⁹ (PDB ID: 7TUZ), LPI-bound GPR55¹⁰ (PDB ID: 8ZX4), 15R-methyl-PGD2-bound DP2¹¹ (PDB ID: 7M8W), 9(S)-HODE-bound GPR132¹² (PDB ID: 8HQN), and DCA-3P-bound MRGPRX4¹³ (PDB ID: 8K4S). **b**, Cross-section representation of EPA-bound FFA4⁴ (PDB ID: 8ID9), ONO-9780307-bound LPA1¹⁴ (PDB ID: 7TD0), S1P-bound S1P1¹⁵ (PDB ID: 7VIE), S1P-bound S1P2¹⁶ (PDB ID: 7T6B), S1P-bound S1P3¹⁷ (PDB ID: 7EW3), Siponimod-bound S1P5¹⁸ (PDB ID: 7EW1), CP55940-bound CB1¹⁹ (PDB ID: 7WV9), AM12033-bound CB2²⁰ (PDB ID: 6KPF), LPC-bound GPR119²¹ (PDB ID: 7XZ5), 3-OH-C12-bound GPR84²² (PDB ID: 8J18), LTB4-bound BLT1²³ (PDB ID: 7VKT), PGE2-bound EP2²⁴ (PDB ID: 7CX2), PGE2-bound EP3²⁵ (PDB ID: 8GDC), PGE2-bound EP4²⁶ (PDB ID: 7D7M), PGF2 α -bound FP2²⁷ (PDB ID: 8IUK), MRE-269-bound IP²⁸ (PDB ID: 8X79), Cloprosetnol-bound TP²⁹ (PDB ID: 8XJN), and oleic acid-bound GPR3³⁰ (PDB ID: 8WW2). In all panels, carbon, oxygen, nitrogen, and phosphorus atoms of each ligand are colored in gray, red, blue, and yellow, respectively. The stars in the middle of each receptor represent the position of the residues at 6.48 in the BW numbering. The dashed lines represent the distance between the residues at 6.48 and the closest atoms of a polar group (**a**) or hydrocarbon chain (**b**) of each ligand.

3. The authors propose that GLPG0974 acts as an allosteric antagonist by occupying a different site than the orthosteric agonist. Did the authors measure the inhibition efficacy of GLPG0974 in receptors stimulated by different agonists? I am particularly interested in the potential probe dependence of GLPG0974 as an allosteric antagonist, similar to the case presented in DOI: 10.1038/s41392-023-01625-y. Please explore this feature further.

Our original manuscript demonstrated that GLPG0974 acts as an allosteric antagonist by binding to allosteric site 1, which borders but does not overlap the orthosteric pocket. This binding induces a conformational shift of Y90^{3.33} toward the orthosteric site, which subsequently collapses the orthosteric pocket and inhibits the binding of orthosteric agonists such as SCFAs and TUG-1375. This mechanism of GLPG0974 differs substantially from the mechanism proposed for most negative allosteric modulators, such as compound 9n in HCA2³¹, as noted by Reviewer #1. Given this mechanism, we anticipated that GLPG0974 would competitively block orthosteric agonist binding but would not interfere with 4-CMTB binding, which binds to the lateral surface of TM6-TM7, distal from the orthosteric site. To confirm this mechanism, we assessed the inhibitory activity of GLPG0974 against four agonists: propionate, butyrate, TUG-1375, and 4-CMTB. As expected, GLPG0974 inhibited the activity of the orthosteric agonists propionate, butyrate, and TUG-1375 but did not inhibit receptor activation induced by 4-CMTB (Response Fig. 4a-d). These results support our hypothesis and confirm the mechanism of GLPG0974.

Response Fig. 4 Antagonist activity of GLPG0974

a-d, Antagonist activity of GLPG0974 against propionate (**a**), butyrate (**b**), TUG-1375 (**c**), and 4-CMTB (**d**) evaluated using the TGF α shedding assay. Symbols and error bars denote mean and SEM, respectively, from three independent experiments.

4. Given the unexpected mode of action of GLPG0974, I suggest conducting additional experiments to clarify this novel antagonistic behavior. In Extended Data Figure 5, the author validated the ligand-binding poses using MD simulation, I recommend extending the MD simulations to at least 500 ns to ensure a more robust evaluation of ligand-receptor interactions.

Following Reviewer #1's suggestion, we extended our MD simulations from 100 ns to 500 ns to better assess the binding pose of GLPG0974 (**Response Fig. 5a-h / Supplementary Fig. 5c-j**). Additionally, as recommended by another reviewer, we performed simulations that accounted for the protonation states of both K65^{2.60} and E166^{ECL2} (see also **Response Fig. 7**). The results showed that the conformational changes of GLPG0974 observed during the initial 100 ns simulation remained stable throughout the extended simulation, supporting our conclusion that pose 1 is the most plausible binding pose.

Response Fig. 5 (= Supplementary Fig. 5c-j) Computational characterization of antagonist binding. **a-d**, MD simulations of GLPG0974 in inactive FFA2 with different poses and protonation conditions. The non-transparent sticks represent the GLPG0974 structure in the cryo-EM model, while the semi-transparent sticks depict the GLPG0974 structure in the last snapshots taken at 500 ns from each of the three simulation replicates.

e-h, MD trajectories for three independent simulations, showing the RMSD (Å) of GLPG0974 with respect to the cryo-EM model. The thin lines represent unsmoothed values, while the thick lines represent a moving average using a smoothing window of 1 ns.

5. The manuscript highlights the probe dependence of 4-CMTB, which enhances the

potency of SCFAs but not synthetic agonists like TUG-1375. Have the authors tested 4-CMTB's effects with other classes of ligands beyond SCFAs and TUG-1375 to confirm this observation?

Although we attempted to further evaluate the generality of 4-CMTB's probe dependence, we encountered challenges due to the absence of commercially available synthetic orthosteric agonists for FFA2. A thorough review of databases, including the IUPHAR/BPS Guide to Pharmacology and GPCRdb, confirmed that no other synthetic orthosteric FFA2 agonists are on the market. This limited availability of synthetic agonists restricts our ability to characterize the probe dependence of 4-CMTB. Future development and accessibility of synthetic FFA2 agonists with varying potencies would enable a more comprehensive examination of this phenomenon.

6. The binding of 4-CMTB to the outer surface of TM6-TM7 and its stabilization of this conformation is intriguing. Does this PAM effect involve any classical motifs typically associated with receptor activation? For instance, an allosteric modulator of the beta-2 adrenergic receptor binds outside of TM5-TM6 and regulates the P-I-F motif. Please provide a comparative discussion on this point.

As briefly described in the original manuscript, our FFA2 structures indicate that 4-CMTB influences the dynamics of the Na⁺ binding pocket and the NPxxY/DPxxF motif. In FFA2, the NPxxY motif is replaced by DPxxF, with D269^{7.49} forming part of the Na⁺ binding pocket alongside D55^{2.50}, S96^{3.39}, and N265^{7.45} in the inactive state, thereby stabilizing the receptor (Response Fig. 6a / Fig. 6e in the manuscript). Additionally, D269^{7.49} interacts with N230^{6.43}, further maintaining the inactive conformation (Response Fig. 6a / Fig. 6e in the manuscript). Upon receptor activation, the D269^{7.49}-N230^{6.43} interaction breaks, and N230^{6.43} reorients toward the lipid bilayer as TM6 undergoes a clockwise rotation (Response Fig. 6a / Fig. 6e, f, h in the manuscript). In this active state, 4-CMTB forms a hydrogen bond with N230^{6.43}, preventing the receptor from returning to the inactive conformation (Response Fig. 6a / Fig. 5c, g in the manuscript). Since D269^{7.49} is a key component of both the Na⁺ binding pocket and the DPxxF motif, 4-CMTB likely affect the dynamics of these regions via N230^{6.43}. We clarified this point by editing the manuscript and adding Response Fig. 6a as Supplementary Fig. 7j.

In addition, to better contextualize the binding of 4-CMTB, we compared 4-CMTB and other allosteric modulators regarding their effects on the GPCR activation motifs. For example, as noted by the reviewer, the negative allosteric modulator AS408³², which targets β2AR, binds to the external surface of TMs 3 and 5 and inhibits structural changes in the PIF motif—a key element for GPCR activation (Response Fig. 6b). Similarly, the positive allosteric modulator ZCZ011¹⁹, which targets CB1, binds to the external surface of TMs 2, 3, and 4, contributing to the destabilization of the Na⁺ pocket and stabilizing the active state of CB1 (Response Fig. 6c). We edited the manuscript, added Response Fig. 6b and 6c as Supplementary Figs. 13e and 13f, and described the diversity in binding sites and

mechanisms of these allosteric modulators binding to the outer surface of the TM region.

Response Fig. 6 Comparison of allosteric modulators

a (Supplementary Fig. 7j), Superimposed image of active FFA2 bound to 4-CMTB (purple, cyan) and inactive FFA2 (beige).

b (Supplementary Fig. 13e), Superimposed image of active β 2AR (active) (PDB ID: 4LDO) and inactive β 2AR bound to AS408 (orange, purple) (PDB ID: 6OBA).

c (Supplementary Fig. 13f), Superimposed image of active CB1 bound to ZCZ011 (pink, orange) (PDB ID: 7WV9) and inactive CB1 (light blue) (PDB ID: 5U09).

a-c, Hydrogen bonds are represented as dashed black lines.

References for Reviewer #1

- 1 Xu, Z. *et al.* Structural basis of sphingosine-1-phosphate receptor 1 activation and biased agonism. *Nat Chem Biol* **18**, 281-288, doi:10.1038/s41589-021-00930-3 (2022).
- 2 Mao, C. *et al.* Unsaturated bond recognition leads to biased signal in a fatty acid receptor. *Science* **380**, eadd6220, doi:10.1126/science.add6220 (2023).
- 3 Kumari, P., Inoue, A., Chapman, K., Lian, P. & Rosenbaum, D. M. Molecular mechanism of fatty acid activation of FFAR1. *Proc Natl Acad Sci U S A* **120**, e2219569120, doi:10.1073/pnas.2219569120 (2023).
- 4 Li, F. *et al.* Molecular recognition and activation mechanism of short-chain fatty acid receptors FFAR2/3. *Cell Res*, doi:10.1038/s41422-023-00914-z (2024).
- 5 Xia, A. *et al.* Cryo-EM structures of human GPR34 enable the identification of selective antagonists. *Proc Natl Acad Sci U S A* **120**, e2308435120, doi:10.1073/pnas.2308435120 (2023).
- 6 Yin, H. *et al.* Insights into lysophosphatidylserine recognition and Galpha(12/13)-coupling specificity of P2Y10. *Cell Chem Biol* **31**, 1899-1908 e1895, doi:10.1016/j.chembiol.2024.08.005 (2024).
- 7 Liang, J. *et al.* Structural basis of lysophosphatidylserine receptor GPR174 ligand recognition and activation. *Nat Commun* **14**, 1012, doi:10.1038/s41467-023-36575-0 (2023).
- 8 Fan, W. *et al.* Molecular basis for the activation of PAF receptor by PAF. *Cell Rep* **43**, 114422, doi:10.1016/j.celrep.2024.114422 (2024).
- 9 Chen, H., Huang, W. & Li, X. Structures of oxysterol sensor EBI2/GPR183, a key regulator of the immune response. *Structure* **30**, 1016-1024 e1015, doi:10.1016/j.str.2022.04.006 (2022).
- 10 Xia, R. *et al.* Structural insight into GPR55 ligand recognition and G-protein coupling. *Cell Res*, doi:10.1038/s41422-024-01044-w (2024).
- 11 Liu, H. *et al.* Molecular basis for lipid recognition by the prostaglandin D(2) receptor CRTH2. *Proc Natl Acad Sci U S A* **118**, doi:10.1073/pnas.2102813118 (2021).
- 12 Wang, J. L. *et al.* Functional screening and rational design of compounds targeting GPR132 to treat diabetes. *Nat Metab* **5**, 1726-1746, doi:10.1038/s42255-023-00899-4 (2023).
- 13 Yang, J. *et al.* Structure-guided discovery of bile acid derivatives for treating liver diseases without causing itch. *Cell* **187**, 7164-7182 e7118, doi:10.1016/j.cell.2024.10.001 (2024).
- 14 Liu, S. *et al.* Differential activation mechanisms of lipid GPCRs by lysophosphatidic acid and sphingosine 1-phosphate. *Nat Commun* **13**, 731, doi:10.1038/s41467-022-28417-2 (2022).
- 15 Yu, L. *et al.* Structural insights into sphingosine-1-phosphate receptor activation. *Proc Natl Acad Sci U S A* **119**, e2117716119, doi:10.1073/pnas.2117716119 (2022).
- 16 Chen, H. *et al.* Structure of S1PR2-heterotrimeric G(13) signaling complex. *Sci Adv* **8**, eabn0067, doi:10.1126/sciadv.abn0067 (2022).
- 17 Zhao, C. *et al.* Structural insights into sphingosine-1-phosphate recognition and ligand selectivity of S1PR3-Gi signaling complexes. *Cell Res* **32**, 218-221, doi:10.1038/s41422-021-00567-w (2022).
- 18 Yuan, Y. *et al.* Structures of signaling complexes of lipid receptors S1PR1 and S1PR5 reveal mechanisms of activation and drug recognition. *Cell Res* **31**, 1263-1274, doi:10.1038/s41422-021-00566-x (2021).
- 19 Yang, X. *et al.* Molecular mechanism of allosteric modulation for the cannabinoid receptor CB1. *Nat Chem Biol* **18**, 831-840, doi:10.1038/s41589-022-01038-y (2022).
- 20 Hua, T. *et al.* Activation and Signaling Mechanism Revealed by Cannabinoid Receptor-G(i) Complex Structures. *Cell* **180**, 655-665 e618, doi:10.1016/j.cell.2020.01.008 (2020).
- 21 Xu, P. *et al.* Structural identification of lysophosphatidylcholines as activating ligands for orphan receptor GPR119. *Nat Struct Mol Biol* **29**, 863-870, doi:10.1038/s41594-022-00816-5 (2022).
- 22 Liu, H. *et al.* Structural insights into ligand recognition and activation of the medium-chain fatty acid-sensing receptor GPR84. *Nat Commun* **14**, 3271, doi:10.1038/s41467-023-38985-6 (2023).

- 23 Wang, N. *et al.* Structural basis of leukotriene B4 receptor 1 activation. *Nat Commun* **13**, 1156, doi:10.1038/s41467-022-28820-9 (2022).
- 24 Qu, C. *et al.* Ligand recognition, unconventional activation, and G protein coupling of the prostaglandin E(2) receptor EP2 subtype. *Sci Adv* **7**, doi:10.1126/sciadv.abf1268 (2021).
- 25 Huang, S. M. *et al.* Single hormone or synthetic agonist induces G(s)/G(i) coupling selectivity of EP receptors via distinct binding modes and propagating paths. *Proc Natl Acad Sci U S A* **120**, e2216329120, doi:10.1073/pnas.2216329120 (2023).
- 26 Nojima, S. *et al.* Cryo-EM Structure of the Prostaglandin E Receptor EP4 Coupled to G Protein. *Structure* **29**, 252-260 e256, doi:10.1016/j.str.2020.11.007 (2021).
- 27 Wu, C. *et al.* Ligand-induced activation and G protein coupling of prostaglandin F(2alpha) receptor. *Nat Commun* **14**, 2668, doi:10.1038/s41467-023-38411-x (2023).
- 28 Wang, J. J. *et al.* Molecular recognition and activation of the prostacyclin receptor by anti-pulmonary arterial hypertension drugs. *Sci Adv* **10**, eadk5184, doi:10.1126/sciadv.adk5184 (2024).
- 29 Li, X. *et al.* Structural basis for ligand recognition and activation of the prostanoid receptors. *Cell Rep* **43**, 113893, doi:10.1016/j.celrep.2024.113893 (2024).
- 30 Xiong, Y. *et al.* Identification of oleic acid as an endogenous ligand of GPR3. *Cell Res*, doi:10.1038/s41422-024-00932-5 (2024).
- 31 Cheng, L. *et al.* Orthosteric ligand selectivity and allosteric probe dependence at Hydroxycarboxylic acid receptor HCAR2. *Signal Transduct Target Ther* **8**, 364, doi:10.1038/s41392-023-01625-y (2023).
- 32 Liu, X. *et al.* An allosteric modulator binds to a conformational hub in the beta(2) adrenergic receptor. *Nat Chem Biol* **16**, 749-755, doi:10.1038/s41589-020-0549-2 (2020).

Reviewer #2

Two sets of MD simulations were performed to propose and refine structural models and describe atomic interactions of the FFA2 receptor in complex with 1) propionate, an orthosteric agonist, and 2) the orthosteric antagonist GLPG0974.

In the propionate-receptor complex system, the propionate stability was analyzed through 12 independent simulations.

For the GLPG0974-receptor complex system, the protonated and deprotonated state of K652 was performed to define the stability of the two GLPG0974 binding modes. The simulations were performed in POPC lipid membranes.

The CHARMM36M and CHARMM22 force files were used for proteins, lipids, and ions, which are well-established and validated for transmembrane proteins. The CHARMM TIP3P model was used for water, which has been thoroughly validated.

The authors performed multiple independent simulations that enhanced their data's reliability and validated the simulations' sampling.

Points that need to be addressed.

** The figures, figure legends, and figure references from the extended data DO NOT CORRESPOND. Therefore, the paper was not easy to follow. It should be corrected!*

Thank you very much for bringing this issue to our attention. Following revisions to the manuscript and incorporations of the main and supplementary figures in response to all reviewers' comments, we thoroughly reexamined the entire manuscript to ensure that all figure panels and legends accurately correspond to their references in the text. We believe these corrections have improved the clarity and coherence of the paper, making it easier to follow.

** It is unclear why different protocols were used to address the protonation states of the simulated systems. The use of only two protonation states for K652.60 in GLPG0974 might oversimplify the actual range of possible states.*

Regarding the first concern, in the original manuscript, we determined the protonation states of residues in the active FFA2 bound to propionate by visually assessing their surrounding environments. In contrast, in the inactive FFA2 structure bound to GLPG0974, there were several residues near GLPG0974 whose protonation states are difficult to assess visually (e.g., K65, E68, E166). Therefore, we employed the MEAD and Karlsberg programs to calculate the protonation probabilities of these residues. To ensure methodological consistency, we also applied these programs to the active FFA2 structure and confirmed that the

calculated protonation states were essentially consistent with our initial visual assessments.

For the second concern, in the original manuscript, we focused solely on K65^{2.60}, as it exhibited differences in calculated protonation states between pose 1 (deprotonated) and pose 2 (protonated). Accordingly, we conducted simulations considering both the protonated and deprotonated states of K65^{2.60}.

To strengthen our analysis, we considered the differences in protonation states between GLPG0974-bound FFA2 and apo FFA2 to account for changes in protonation states during the ligand-binding process. This analysis identified an additional residue, E166^{ECL2}, with differing protonation states. Consequently, we performed simulations under the following three conditions to assess the validity of pose 1 and pose 2:

1. K65^{2.60} protonated / E166^{ECL2} protonated (K65p_E166p): GLPG0974-bound (pose 1) state
2. K65^{2.60} deprotonated / E166^{ECL2} protonated (K65d_E166p): GLPG0974-bound (pose 2) state
3. K65^{2.60} protonated / E166^{ECL2} deprotonated (K65p_E166d): Apo state

Note that, as recommended by another reviewer, these simulations were extended to 500 ns.

The results showed that under conditions (1) K65p_E166p and (2) K65d_E166p, the findings matched our initial 100 ns simulations (Response Fig. 7a, b, d, e / Supplementary Fig. 5c-j). Specifically, GLPG0974 remained stable in its initial position in pose 1 throughout the 500 ns simulation, whereas in pose 2, GLPG0974 exhibited significant dynamic changes. Interestingly, under the newly added condition (3) K65p_E166d, GLPG0974 remained stably bound in both pose 1 and pose 2, contrary to expectations (Response Fig. 7c, f). This stability resulted from the deprotonated E166^{ECL2} forming stable hydrogen bonds with R180^{5.39} and R255^{7.35}, significantly reducing the flexibility of ECL2 (Response Fig. 7g-i).

However, in the inactive FFA2 structure bound to GLPG0974 that we determined, E166^{ECL2} and R180^{5.39} are 5.2 Å apart and do not form a hydrogen bond (Response Fig. 7j). Moreover, previous studies have shown that GLPG0974 binding is maintained in an FFA2 mutant where R180^{5.39} is substituted with alanine³³. These findings indicate that GLPG0974 can bind to FFA2 even when E166^{ECL2} and R180^{5.39} do not form a hydrogen bond. Based on these results, we conclude that E166^{ECL2} is protonated and that pose 1 represents the most plausible binding mode for GLPG0974.

Response Fig. 7 Computational characterization of antagonist binding.

a-f, MD simulations of GLPG0974 in inactive FFA2 with different poses and protonation conditions. The non-transparent sticks represent the GLPG0974 structure in the cryo-EM model, while the semi-transparent sticks depict the GLPG0974 structure in the last snapshots taken at 500 ns from each of the three simulation replicates. MD trajectories for three independent simulations, showing the RMSD (Å) of GLPG0974 with respect to the cryo-EM model. The thin lines represent unsmoothed values, while the thick lines represent a moving average using a smoothing window of 1 ns.

- g**, MD trajectories from three independent simulations, showing the distance (Å) of E166^{ECL2} and R180^{5.39}.
- h**, MD trajectories from three independent simulations, showing the distance (Å) of E166^{ECL2} and R255^{7.35}.
- i**, MD trajectories from three independent simulations, showing the RMSD (Å) of ECL2 with respect to the cryo-EM model.
- j**, Cryo-EM density map and models for E166^{ECL2}, R180^{5.39}, and R255^{7.35}.

** The authors found a key water molecule forming a water bridge between R255 and the GLPG0974 ligand. What happened to the waters through the simulated systems inside the protein cavities? Do the active and inactive conformation systems contain different water arrangements and dynamics in the protein cavities/channel?*

We appreciate this suggestion and comprehensively analyzed the distribution of water molecules within the protein cavities of both active and inactive FFA2 through MD simulations. This analysis identified six distinct water clusters within the central region of the transmembrane helices (Response Fig. 8).

As shown in Response Fig. 8a-f, five of the six clusters were present in both active and inactive states (Response Fig. 8a-e), while one cluster near residues N25^{1.50}, T52^{2.47}, and D55^{2.50} was observed only in the active state (Response Fig. 8f). Intriguingly, this site is reported as a conserved water-binding site in both active and inactive states across Class A GPCRs³⁴. However, in FFA2, the unique position of TM7 in the inactive state causes F273^{7.53} on TM7 to occupy the binding site. This site becomes available for water molecules only after receptor activation and the accompanying conformational change in F273^{7.53}. Thus, unlike other Class A GPCRs, this water cluster in FFA2 may specifically stabilize the receptor in its active state. We added Response Fig. 8 as Supplementary Fig. 11 and described this water distribution analysis in the discussion section of the main manuscript.

Response Fig. 8 (Supplementary Fig. 11) Water molecules in MD simulations
a–f, Superimposed simulation frames sampled at every 50 ns from 50 to 500 ns in three representative simulations of the active state receptor and three representative simulations of the inactive state receptor. Water clusters observed between A258^{7.38} and S262^{7.42} (**a**), T97^{3.40} and N239^{6.52} (**b**), S104^{3.47}, T195^{5.54}, and T227^{6.40} (**c**), D55^{2.50}, S96^{3.39}, and N265^{7.45} (**d**), N265^{7.45} and N269^{7.49} (**e**), and N25^{1.50}, T52^{2.47}, D55^{2.50}, and F273^{7.53} (**f**) are enlarged in each dashed rectangle. Water molecules observed in active state simulations are represented as magenta spheres, while those observed in inactive state simulations are represented as green spheres. The receptor in the active and inactive states is depicted as semi-transparent and colored in purple and beige, respectively. The black arrow in (**f**) represents the movement of F273^{7.53} upon receptor activation.

Reference for Reviewer #2

- 33 Sergeev, E. *et al.* Non-equivalence of Key Positively Charged Residues of the Free Fatty Acid 2 Receptor in the Recognition and Function of Agonist Versus Antagonist Ligands. *J Biol Chem* **291**, 303-317, doi:10.1074/jbc.M115.687939 (2016).
- 34 Venkatakrisnan, A. J. *et al.* Diverse GPCRs exhibit conserved water networks for stabilization and activation. *Proc Natl Acad Sci U S A* **116**, 3288-3293, doi:10.1073/pnas.1809251116 (2019).

Reviewer #3

In this study, Mai Kugawa et al., determined the cryo-EM structures of active FFA2-Gi complex with the orthostatic agonist orthosteric agonist TUG-1375 and the PAM 4-CMTB, and the inactive FFA2 structure bound to the antagonist GLPG0974. By comparing these structures, together with MD simulations and functional studies, these findings shed light on the ligand recognition and activation of orthosteric / allosteric ligands. It also uncovers the binding site and mechanisms of clinically relevant antagonist GLPG0974. In general, this is an informative paper with well organised presentation of the data. The cryo-EM maps and preliminary PDB reports reflects the good quality of the structure deposited.

We greatly appreciate Reviewer #3's positive feedback, specifically their recognition of our work as "informative" and their acknowledgment of the "well-organized presentation" and "good quality of the structure deposited" in our study.

Minor comments as below:

1. Have the constructs of FFA2 used in this study been validated? Do they make pharmacological sense? These data should be included. Or the references should be given if they have been validated in previous studies.

We appreciate the reviewer's request for additional characterization of the receptor constructs used in our structural studies.

First, to determine the active FFA2 structure, we utilized an FFA2 construct lacking the C-terminal five residues, designated as FFA2 (1–325). We evaluated the pharmacological activity of this construct using the TGF α shedding assay. As shown in **Response Fig. 9a, the FFA2 (1–325) construct exhibited pharmacological properties comparable to those of the wild-type, full-length receptor across all ligands tested (**Response Fig. 9a / Supplementary Fig. 1b in the revised manuscript**). Next, for the inactive FFA2 structure, we used a BRIL-fused FFA2 construct, where BRIL was fused to intracellular loop 3 (ICL3) between TMs 5 and 6. The fused BRIL physically blocks G-protein binding, preventing us from assessing whether the BRIL fusion significantly affects the receptor's structure by measuring pharmacological activity using the TGF α shedding assay. Instead, we removed BRIL from the cryo-EM structure of BRIL-fused FFA2 and performed MD simulations. As shown in **Response Fig. 9b, c**, three independent simulations demonstrated that the receptor's conformation remains largely unchanged after the removal of BRIL, suggesting that**

the BRIL fusion to ICL3 does not significantly affect the conformation of FFA2 (Response Fig. 9b, c).

Response Fig. 9 Evaluations of the constructs used in cryo-EM analysis

a, (Supplementary Fig. 1b) Agonist activity of propionate (a), butyrate (b), TUG-1375 (c) and 4-CMTB (d) against WT FFA2 (black) and FFA2 (1-325) (purple) construct evaluated by the TGF α shedding assay. Symbols and error bars denote mean and SEM, respectively, from three independent experiments.

b, Superimposed image of the cryo-EM model (beige) and MD snapshots (gray).

c, MD trajectories for three independent simulations showing the RMSD (Å) of the whole protein with respect to the cryo-EM model.

2. Can the PAM 4-CMTB modulate the synthetic orthosteric agonist TUG-1375? If not, why did the authors choose this combination? And will the binding of PAM cause any effect on the orthosteric ligand binding? It is worthwhile to further discuss these in the discussion part.

We appreciate the reviewer for giving us the opportunity to clarify this matter. We used both TUG-1375 and 4-CMTB to obtain the binding site and pose of 4-CMTB because 4-CMTB alone could not sufficiently stabilize the FFA2-Gi complex. As shown in Response Fig. 10a and 10b, the efficiency of FFA2-Gi protein complex formation was significantly lower when only 4-CMTB was added compared to when both TUG-1375 and 4-CMTB were present (Response Fig. 10a, b).

Regarding the second question, we do not think that 4-CMTB binding significantly affects the binding of TUG-1375. Indeed, structural comparisons between our TUG-1375/4-CMTB-bound FFA2 structure and the recently reported TUG-1375-bound FFA2 structure⁴ (PDB ID: 8J22) reveal that the binding sites in both structures are remarkably similar (Response Fig. 10c). This supports our proposed mechanism of probe dependence by 4-CMTB: 4-CMTB acts as a PAM not by affecting the orthosteric binding pocket but by influencing the receptor's equilibrium between active and inactive states. Therefore, 4-CMTB cannot exert a further PAM effect on an already potent agonist. We added Response Fig. 10c as Supplementary Fig. 13g, h and edited the discussion section of the main manuscript accordingly.

Response Fig. 10 Structural analysis of 4-CMTB-bound structure

a, Fluorescence-detection size-exclusion chromatography (FSEC) traces of FFA2 alone (black) and FFA2-Gi complex (magenta) bound to 4-CMTB.

b, FSEC traces of FFA2 alone (black) and FFA2-Gi complex (magenta) bound to 4-CMTB and TUG-1375.

c, (Supplementary Fig. 13g, h), Comparison of the orthosteric pocket of FFA2 in complex with both TUG-1375 and 4-CMTB (purple, green) and with only TUG-1375 (dark gray, green) (PDB ID: 8J22).

3. The side chain rotamer of most key residues discussed in this study were well supported by the cryo-EM map density. However, the density of residues 65, 242, 255 of the inactive structure are not so well resolved, and there might be different ways of modelling. Therefore, I am not so convinced of the related activation / inactivation mechanisms presented in Fig. 6.

We appreciate the reviewer's thorough evaluation of our structures and invaluable comments on the modeling of K65^{2.60}, H242^{6.55}, and R255^{7.35}. Regarding K65^{2.60} and R255^{7.35}, we acknowledge that, although corresponding densities were present, the limited resolution of the cryo-EM map introduces ambiguity about the conformations of their side chains. We carefully analyzed the density and re-refined the conformations of these two residues (Response Fig. 11a, b).

Regarding H242^{6.55}, as pointed out by Reviewer #3, a clear cryo-EM density map for its side chain was also lacking. However, other conformations we tested showed a poorer fit to the density. Therefore, we conducted three additional MD simulations and found that the current conformer is energetically stable, supporting the originally modeled conformation (Response Fig. 11c-e). Nonetheless, we acknowledge that this evidence provides limited support; therefore, we have moderated our statements about receptor activation, especially concerning the process involving H242^{6.55}.

Response Fig. 11 improvement of structural modeling

- a,b**, Cryo-EM density map and models in initial submission (light blue) and after re-refinement (beige), focused on K65^{2,60} and R255^{7,35}.
- c,d**, Cryo-EM density map and model in initial submission (light blue) and MD snapshots (gray), focused on H242^{6,55}.
- e**, MD trajectories from three independent simulations, showing the RMSD (Å) of H242^{6,55} with respect to the cryo-EM model.

4. *In the extended Fig. 7a, why is the agonist TUG-1375 present in an inactive structure?*

Thank you for pointing this out. Originally, we superimposed TUG-1375 onto the inactive FFA2 structure in **Extended Data Fig. 7a**, but we agree that this was confusing. To avoid misunderstanding, we have removed TUG-1375 from **Extended Data Fig. 7a** (**now renamed Supplementary Fig. 7a**).

5. *As there are relevant structures published more than half year ago (Ref 43, 44), particularly the FFA2-Gi active structures bound to the same ligand TUG-1375 or endogenous SCFA acetate, which directly related to this study. I would suggest the authors to fully discuss these structures.*

We appreciate this suggestion and compared our TUG-1375/4-CMTB-bound FFA2 structure with four recently reported FFA2 structures bound to TUG-1375, acetate, or butyrate^{4,35,36} (**Response Fig. 12a**).

The five FFA2 structures align well with one another (**Response Fig. 12a**), including the ligand binding pockets (**Response Fig. 12b–d**). Notably, key residues in the orthosteric binding pocket that interact with TUG-1375—Y90^{3,33}, R180^{5,39}, Y238^{6,51}, H242^{6,55}, and R255^{7,35}—exhibit nearly identical conformations. Moreover, although there are small structural deviations, the carboxyl groups of each ligand are positioned nearly identically, as suggested by our MD simulations of propionate-bound FFA2 in the original manuscript (**Supplementary Fig. 4b**). We added **Response Fig. 12a–d as Supplementary Fig. 13g–j** and edited the discussion section of the main manuscript accordingly.

Response Fig. 12 (Supplementary Fig. 13g-j) Comparison of FFA2 structures

a, Superimposed image of FFA2 structure in this study and three other reported FFA2 structures (PDB ID: 8J22, 8J24, 8T3S). **b**, Superimposed image of TUG-1375/4-CMTB-bound FFA2 (this study) and TUG-1375-bound FFA2 (PDB ID: 8J22), focused on the TUG-1375-binding site. **c**, Superimposed image of TUG-1375/4-CMTB-bound FFA2 (this study) and acetate-bound FFA2 (PDB ID: 8J24), focused on the TUG-1375-binding site. **d**, TUG-1375/4-CMTB-bound FFA2 (this study) and butyrate-bound FFA2 (PDB ID: 8T3S), focused on the TUG-1375-binding site.

References for Reviewer #3

- 4 Li, F. *et al.* Molecular recognition and activation mechanism of short-chain fatty acid receptors FFAR2/3. *Cell Res*, doi:10.1038/s41422-023-00914-z (2024).
- 35 Zhang, X. *et al.* Structural basis for the ligand recognition and signaling of free fatty acid receptors. *Sci Adv* **10**, eadj2384, doi:10.1126/sciadv.adj2384 (2024).
- 36 Ke, Y. *et al.* Structural insights into endogenous ligand selectivity and activation mechanisms of FFAR1 and FFAR2. *Cell Rep* **43**, 115024, doi:10.1016/j.celrep.2024.115024 (2024).